# MEAN-VARIANCE EFFICIENT REINFORCEMENT LEARNING BY EXPECTED QUADRATIC UTILITY MAXIMIZATION

## ABSTRACT

In reinforcement learning (RL) for sequential decision making under uncertainty, existing methods proposed for considering *mean-variance* (MV) trade-off suffer from computational difficulties in computation of the gradient of the variance term. In this paper, we aim to obtain MV-efficient policies that achieve Pareto efficiency regarding MV trade-off. To achieve this purpose, we train an agent to maximize the expected quadratic utility function, in which the maximizer corresponds to the Pareto efficient policy. Our approach does not suffer from the computational difficulties because it does not include gradient estimation of the variance. In experiments, we confirm the effectiveness of our proposed methods.

## 1 INTRODUCTION

*Reinforcement learning* (RL) trains intelligent agents to solve sequential decision-making problems (Puterman, 1994; Sutton & Barto, 1998). While a typical objective is to maximize the expected cumulative reward, *risk-aware* RL has recently attracted much attention in real-world applications, such as finance and robotics (Geibel & Wysotzki, 2005; Garcıa & Fernández, 2015). Various criteria have been proposed to capture a risk, such as Value at Risk (Chow & Ghavamzadeh, 2014; Chow et al., 2017) and variance (Markowitz, 1952; Luenberger et al., 1997; Tamar et al., 2012; Prashanth & Ghavamzadeh, 2013). Among them, we consider the *mean-variance RL* (MVRL) methods that attempt to train an agent while controlling the mean-variance (MV) trade-off (Tamar et al., 2012; Prashanth & Ghavamzadeh, 2013; 2016; Xie et al., 2018; Bisi et al., 2020; Zhang et al., 2021b).

Existing MVRL methods (Tamar et al., 2012; Prashanth & Ghavamzadeh, 2013; 2016; Xie et al., 2018), typically maximize the expected cumulative reward while keeping the variance of the cumulative reward at a certain level or, equivalently, minimize the variance while keeping the expected cumulative reward at a certain level. These MVRL methods simultaneously estimate the expected reward or variance while training an agent and solve the constrained optimization problem relaxed by penalized methods. These studies have reported that RL-based methods suffer from high computational difficulty owing to the *double sampling issue* (Section 3) when approximating the gradient of the variance term (Tamar et al., 2012; Prashanth & Ghavamzadeh, 2013; 2016). To avoid this, Tamar et al. (2012) and Prashanth & Ghavamzadeh (2013) proposed multi-time-scale stochastic optimization. Further, Xie et al. (2018) proposed a method based on the Legendre-Fenchel duality (Boyd & Vandenberghe, 2004). Although these methods avoid the double sampling issue, as we experimentally report in Figure 2 of Section 6, there still remains the difficulty in training a policy.

To avoid these difficulties, this paper considers another approach for MVRL by focusing on obtaining a policy that is located on the Pareto efficient frontier in the sense of MV trade-off; that is, we cannot increase the expected reward without increasing the variance and decrease the variance without decreasing the expected reward (Section 2.2). To achieve this purpose, we propose training a RL agent by direct *expected quadratic utility maximization* (EQUMRL) with a policy gradient method (Williams, 1988; 1992; Sutton et al., 2000; Baxter & Bartlett, 2001). We show that the maximizer of the objective in EQUMRL is Pareto efficient in the sense of MV trade-off.

As an important property, EQUMRL does not suffer from the double-sampling problem because it does not include the variance estimation, which is a cause of the problem. In conventional methods, we can also obtain an MV-efficient policy when we succeed in solving the constraint problem. However,

as discussed in the work and shown in our experiments, such as Figure 2, those methods do not perform well due to the difficulty in computation. Compared with them, EQUMRL is computationally friendly and experimentally returns more MV-efficient policies. In addition, the EQUMRL is well suited to financial applications, such as portfolio management because in economic theory, using MV portfolio is justified as a maximizer of the expected quadratic utility function (Markowitz, 1952; Luenberger et al., 1997). We introduce brief survey on expected quadratic utility maximization and finance in Appendix A.We list the following three advantages to the EQUMRL approach:

**(i)** our proposed EQUMRL is able to learn Pareto efficient policies and has plenty of interpretations from various perspectives (Section 4.3);

**(ii)** our proposed EQUMRL does not suffer from the double sampling issue by avoiding explicit approximation of the variance (Section 4);

**(iii)** we experimentally show that our proposed EQUMRL returns more Pareto efficient policies than existing methods (Section 6).

In the following sections, we first formulate the problem setting in Sections 2–3. Then, we propose the main algorithms in Section 4. Finally, we investigate the empirical effectiveness in Section 6.

## 2   PROBLEM SETTING

We consider a standard setting of RL, where an agent interacts with an unfamiliar, dynamic, and stochastic environment modeled by a Markov decision process (MDP) in discrete time. We define an MDP as a tuple $(\mathcal{S}, \mathcal{A}, r, P, P_0)$, where $\mathcal{S}$ is a set of states, $\mathcal{A}$ is a set of actions, $r : \mathcal{S} \times \mathcal{A} \to \mathbb{R}$ is a stochastic reward function with finite mean and variance, $P : \mathcal{S} \times \mathcal{S} \times \mathcal{A} \to [0, 1]$ is a transition kernel, and $P_0 : \mathcal{S} \to [0, 1]$ is an initial state distribution. The initial state $\mathcal{S}_1$ is sampled from $P_0$. Let $\pi_\theta : \mathcal{A} \times \mathcal{S} \to [0, 1]$ be a parameterized stochastic policy mapping states to actions, where $\theta$ is a tunable parameter, and we denote the parameter space by $\Theta$. At time step $t$, an agent chooses an action $A_t$ following a policy $\pi_\theta(\cdot \mid S_t)$. We assume that the policy $\pi_\theta$ is differentiable with respect to $\theta$; that is, $\frac{\partial \pi_\theta(s|a)}{\partial \theta}$ exists.

Let us define the expected cumulative reward from time step $t$ to $u$ as $\mathbb{E}_{\pi_\theta}[G_{t:u}]$, where $G_{t:u} = \sum_{i=0}^{u-t} \gamma^i r(S_{t+i}, A_{t+i})$, $\gamma \in (0, 1]$ is a discount factor and $\mathbb{E}_{\pi_\theta}$ denotes the expectation operator over a policy $\pi_\theta$, and $S_1$ is generated from $P_0$. When $\gamma = 1$, to ensure that the cumulative reward is well-defined, we usually assume that all policies are proper (Bertsekas & Tsitsiklis, 1995); that is, for any policy $\pi_\theta$, an agent goes to a recurrent state $S^*$ with probability 1, and obtains 0 reward after passing the recurrent state $S^*$ at a stopping time $\tau$. This finite horizon setting is called *episodic* MDPs (Puterman, 1994). For brevity, we denote $G_{t:u}$ as $G$ when there is no ambiguity.

In this paper, we consider the trade-off between the mean and variance of the reward. Let us denote the variance of a random variable $W$ under $\pi_\theta$ by $\mathbb{V}_{\pi_\theta}(W)$. Note that $\mathbb{E}_{\pi_\theta}[G]$ and $\mathbb{V}_{\pi_\theta}(G)$ are finite because $r(S_t, A_t)$ has finite mean and variance.

### 2.1   TRAJECTORY VARIANCE PERSPECTIVE.

A direction of MVRL is to consider the trajectory mean $\mathbb{E}_{\pi_\theta}[G_{t:u}]$ and trajectory variance $\mathbb{V}_{\pi_\theta}(G_{t:u})$:

$$\mathbb{E}_{\pi_\theta}[G] \stackrel{\text{trade-off}}{\Longleftrightarrow} \mathbb{V}_{\pi_\theta}(G).$$

A typical method for considering the MV trade-off is to train a policy under some constrains. Tamar et al. (2012), Prashanth & Ghavamzadeh (2013), and Xie et al. (2018) formulated MVRL as

$$\max_{\theta \in \Theta} \mathbb{E}_{\pi_\theta}[G] \quad \text{s.t.} \ \mathbb{V}_{\pi_\theta}(G) = \eta.$$

We call an algorithm based on this formulation a trajectory MV-controlled RL. In the trajectory MV-controlled RL, the goal is to maximize the expected cumulative reward, while controlling the trajectory-variance at a certain level. To be more precise, their actual constraint condition is $\mathbb{V}_{\pi_\theta}(G) \leq \eta$. However, if $\mathbb{V}_{\pi_\theta}(G) = \eta$ is feasible, the optimizer satisfies the equality in applications, where we need to consider MV trade-off, such as a financial portfolio management. Therefore, we only consider an equality constraint. To solve this problem, Tamar et al. (2012), Prashanth & Ghavamzadeh (2013), and Xie et al. (2018) considered a penalized method defined as

$\max_{\theta \in \Theta} \mathbb{E}_{\pi_\theta}[G] - \delta g\big(\mathbb{V}_{\pi_\theta}(G) - \eta\big)$, where $\delta > 0$ is a constant and $g : \mathbb{R} \to \mathbb{R}$ is a penalty function, such as $g(x) = x$ or $g(x) = x^2$. See Remark 1 for more details.

## 2.2 MV-EFFICIENT POLICY.

To consider the MV trade-off avoiding the computational difficulty, this paper aims to train a Pareto efficient policy in the sense of the MV trade-off, where we cannot increase the mean (resp. decrease the variance) without increasing the variance (resp. decreasing the mean). Following the existing literature mainly in economics, finance, and operations research (Luenberger et al., 1997), we define a trajectory MV-efficient policy as a policy $\pi_\theta$ such that there is no other policy with $\theta' \in \Theta$, where $\mathbb{E}_{\pi_\theta}[G] \leq \mathbb{E}_{\pi_{\theta'}}[G]$ and $\mathbb{V}_{\pi_\theta}[G] > \mathbb{V}_{\pi_{\theta'}}[G]$, or $\mathbb{V}_{\pi_\theta}[G] \geq \mathbb{V}_{\pi_{\theta'}}[G]$ and $\mathbb{E}_{\pi_\theta}[G] < \mathbb{E}_{\pi_{\theta'}}[G]$. Efficient frontier is defined as a set of the MV-efficient policies. From the definition of the MV-controlled policies, the trained policies belong to the efficient frontier.

## 3 POLICY GRADIENT AND DOUBLE SAMPLING ISSUE

In this paper, we consider training a policy by policy gradient methods. In MVRL, we usually require the gradients of $\mathbb{E}_{\pi_\theta}[G]$ and $\mathbb{V}_{\pi_\theta}(G) = \mathbb{E}\big[(G - \mathbb{E}_{\pi_\theta}[G])^2\big]$. Tamar et al. (2012) and Prashanth & Ghavamzadeh (2013) show that the gradients of $\mathbb{E}_{\pi_\theta}[G]$ and $\mathbb{E}_{\pi_\theta}[G^2]$ are given as

$$\nabla_\theta \mathbb{E}_{\pi_\theta}[G] = \mathbb{E}_{\pi_\theta}\left[G \sum_{t=1}^{\tau} \nabla_\theta \log \pi_\theta(S_t, A_t)\right], \quad \nabla_\theta \mathbb{E}_{\pi_\theta}[G^2] = \mathbb{E}\left[G^2 \sum_{t=1}^{\tau} \nabla_\theta \log \pi_\theta(S_t, A_t)\right].$$

Because optimizing the policy $\pi_\theta$ directly using the gradients is computationally intractable, we replace them with their unbiased estimators. Suppose that there is a simulator generating a trajectory $k$ with $\{(S_t^k, A_t^k, r(S_t^k, A_t^k))\}_{t=1}^{\tau^k}$, where $\tau^k$ is the stopping time of the trajectory. Then, we can construct unbiased estimators of $\mathbb{E}_{\pi_\theta}[G]$ and $\mathbb{E}_{\pi_\theta}[G^2]$ as follows (Tamar et al., 2012):

$$\widehat{\nabla}_\theta \mathbb{E}_{\pi_\theta}[G] = \widehat{G}^k \sum_{t=1}^{\tau^k} \nabla_\theta \log \pi_\theta(S_t^k, A_t^k) \text{ and } \widehat{\nabla}_\theta \mathbb{E}_{\pi_\theta}[G^2] = \left(\widehat{G}^k\right)^2 \sum_{t=1}^{\tau^k} \nabla_\theta \log \pi_\theta(S_t^k, A_t^k), \quad (1)$$

where $\widehat{G}^k$ is a sample approximation of $\mathbb{E}_{\pi_\theta}[G]$ at the episode $k$. Besides, because $\nabla_\theta\big(\mathbb{E}_{\pi_\theta}[G]\big)^2 = 2\mathbb{E}_{\pi_\theta}[G]\nabla_\theta\mathbb{E}_{\pi_\theta}[G]$, the gradient of the variance is given as $\nabla_\theta \mathbb{V}_{\pi_\theta}(G) = \nabla_\theta\Big(\mathbb{E}_{\pi_\theta}[G^2] - \big(\mathbb{E}_{\pi_\theta}[G]\big)^2\Big) = \mathbb{E}_{\pi_\theta}\big[G^2 \sum_{t=1}^{\tau} \nabla_\theta \log \pi_\theta(S_t, A_t)\big] - 2\mathbb{E}_{\pi_\theta}[G]\nabla_\theta\mathbb{E}_{\pi_\theta}[G]$.

However, obtaining an unbiased estimator of $\nabla_\theta\big(\mathbb{E}_{\pi_\theta}[G]\big)^2 = 2\mathbb{E}_{\pi_\theta}[G]\nabla_\theta\mathbb{E}_{\pi_\theta}[G]$ is difficult because it requires sampling from two different trajectories for approximating $\mathbb{E}_{\pi_\theta}[G]$ and $\nabla_\theta\mathbb{E}_{\pi_\theta}[G]$. This issue is called double sampling issue and makes the optimization problem difficult when we include the variance $\mathbb{V}_{\pi_\theta}(G)$ into the objective function directly. Tamar et al. (2012) and Prashanth & Ghavamzadeh (2013) reported this double sampling issue caused from the gradient estimation of the variance and proposed solutions based on multi-time-scale stochastic optimization. Recall that they consider the penalized objective function $\max_{\theta \in \Theta} \mathbb{E}_{\pi_\theta}[G] - \delta g\big(\mathbb{V}_{\pi_\theta}(G) - \eta\big)$ to obtain MV-controlled policies, where the double sampling issue caused by the gradient of $\delta g\big(\mathbb{V}_{\pi_\theta}(G) - \eta\big)$.

## 4 EQUMRL WITH TRAJECTORY VARIANCE PERSPECTIVE

In social sciences, we often assume that an agent maximizes the expected quadratic utility for considering MV trade-off. Based on this, we propose the EQUMRL for obtaining an MV-efficient policy. For a cumulative reward $G$, we define the quadratic utility function as $u^{\text{trajectory}}(G; \alpha, \beta) = \alpha G - \frac{1}{2}\beta G^2$, where $\alpha > 0$ and $\beta \geq 0$. In EQUMRL, we train a policy by maximizing the expected value of the quadratic utility function,

$$\mathbb{E}_{\pi_\theta}\big[u^{\text{trajectory}}(G; \alpha, \beta)\big] = \alpha\mathbb{E}_{\pi_\theta}[G] - \frac{1}{2}\beta\mathbb{E}_{\pi_\theta}[G^2]. \quad (2)$$

The EQUMRL is agnostic to the learning method, that is, we can implement it with various existing algorithms such as REINFORCE and actor-crtic.

### 4.1 EQUMRL AND MV EFFICIENCY

The expected quadratic utility function $\mathbb{E}_{\pi_\theta}\left[u^{\text{trajectory}}(G;\alpha,\beta)\right]$ is known to be Pareto efficient in the sense of mean and variance when its optimal value satisfies $\mathbb{E}_{\pi_\theta}[G] \le \frac{\alpha}{\beta}$. In order to confirm this, we can decompose the expected quadratic utility as

$$\mathbb{E}_{\pi_\theta}\left[u^{\text{trajectory}}(G;\alpha,\beta)\right] = -\frac{1}{2}\beta\left(\mathbb{E}_{\pi_\theta}[G] - \frac{\alpha}{\beta}\right)^2 + \frac{\alpha^2}{2\beta} - \frac{1}{2}\beta\mathbb{V}_{\pi_\theta}(G). \tag{3}$$

When a policy $\pi \in \Pi$ is the maximizer of the expected quadratic utility, it is equivalent to an MV-efficient policy (Borch, 1969; Baron, 1977; Luenberger et al., 1997). Following (Luenberger et al., 1997, p.237–239), we explain this as follows:

1. Among policies $\pi_\theta$ with a fixed mean $\mathbb{E}_{\pi_\theta}[G] = \mu$, the policy with the lowest variance maximizes the expected quadratic utility function because $\mathbb{E}_{\pi_\theta}\left[u^{\text{trajectory}}(G;\alpha,\beta)\right] = \alpha\mu - \frac{1}{2}\beta\mu^2 - \frac{1}{2}\beta\mathbb{V}_{\pi_\theta}(G)$ is a monotonous decreasing function on $\mathbb{V}_{\pi_\theta}(G)$;

2. Among policies $\pi_\theta$ with a fixed variance $\mathbb{V}_{\pi_\theta}(G) = \sigma^2$ and mean $\mathbb{E}_{\pi_\theta}[G] \le \frac{\alpha}{\beta}$, the policy with the highest mean maximizes the expected quadratic utility function because $\mathbb{E}_{\pi_\theta}\left[u^{\text{trajectory}}(G;\alpha,\beta)\right] = -\frac{1}{2}\beta\left(\mathbb{E}_{\pi_\theta}[G] - \frac{\alpha}{\beta}\right)^2 + \frac{\alpha^2}{2\beta} - \frac{1}{2}\beta\sigma^2$ is a monotonous increasing function on $\mathbb{E}_{\pi_\theta}[G] \le \frac{\alpha}{\beta}$.

Based on the above property, we propose maximizing the expected quadratic utility function in RL; that is, training an agent to directly maximize the expected quadratic utility function for MV control instead of solving a constrained optimization. We call the framework that makes the RL objective function an expected quadratic utility EQUMRL.

Unlike the expected cumulative reward maximization in the standard RL setting, at time $t$, it is desirable to include the past cumulative reward to the state $S_t$ because our objective function depends on it even given $S_t$. Let us consider the objective at time $t$ with the infinite horizon setting:

$$\alpha\mathbb{E}_{\pi_\theta}\left[G_{0:\infty} - \beta G_{0:\infty}^2 | S_0, A_0, r(S_0, A_0), \ldots, S_{t-1}, A_{t-1}, r(S_{t-1}, A_{t-1}), S_t\right]$$
$$= C + \alpha\gamma^t\left\{(1 - 2\beta\gamma^t G_{0:t-1})\mathbb{E}_{\pi_\theta}[G_{t:\infty}|S_t] - \beta\gamma^t\mathbb{E}_{\pi_\theta}[G_{t:\infty}^2|S_t]\right\},$$

where $C$ is a constant and recall that $G_{0:t-1} = \sum_{i=0}^{t-1}\gamma^i r(S_i, A_0)$. Thus, for a better decision-making, we include the past cumulative reward into the state space.

### 4.2 IMPLEMENTATION OF EQUMRL

In this section, we introduce how to train a policy with the EQURL. We defined the objective function of the EQUMRL, and EQUMRL is an agnostic in learning method. As examples, we show an implementation based on REINFORCE (Williams, 1992; Brockman et al., 2016) and Actor-Critic (AC) methods (Williams & Peng, 1991; Mnih et al., 2016). We use unbiased estimators of the gradients defined in (1).

**REINFORCE-based trajectory EQUMRL.** For an episode $k$ with the length $n$, the proposed algorithm replaces $\mathbb{E}_{\pi_\theta}\left[G\right]$ and $\mathbb{E}_{\pi_\theta}\left[G^2\right]$ with the sample approximations $\sum_{t=1}^n \gamma^{t-1}r(S_t, A_t)$ and $\left(\sum_{t=1}^n \gamma^{t-1}r(S_t, A_t)\right)^2$, respectively (Tamar et al., 2012); that is, the unbiased gradients are given as (1). Therefore, for a sample approximation $\widehat{G}^k$ of $\mathbb{E}_{\pi_\theta}\left[G^2\right]$ at the episode $k$, we optimize the policy with ascending the unbiased gradient

$$\widehat{\nabla}\mathbb{E}_{\pi_\theta}\left[u^{\text{trajectory}}(G;\alpha,\beta)\right] = \left(\alpha\widehat{G}^k - \frac{1}{2}\beta\left(\widehat{G}^k\right)^2\right)\sum_{t=1}^n \nabla_\theta \log\pi_\theta(S_t^k, A_t^k).$$

CWe summarize the algorithm as the pseudo-code in Algorithm 1.

Here, we present three advantages of EQUMRL. The first advantage concerns computation. In EQUMRL, we do not suffer from the double sampling issue because the term $(\mathbb{E}_{\pi_\theta}[G])^2$, which causes the problem, is absent, and we must only estimate the gradients of $\mathbb{E}_{\pi_\theta}[G]$ and $\mathbb{E}_{\pi_\theta}\left[G^2\right]$. The second advantage is that it provides a variety of interpretations, as listed in Section 4.3.

The third advantage is the ease of theoretical analysis. For example, referring to the results of Bertsekas & Tsitsiklis (1996), we derive the following result on the convergence of the gradient, which can be applied to simple policy gradient algorithms, not only our proposed REINFORCE-based algorithm.

---

**Algorithm 1** REINFORCE-based EQUMRL

Initialize the policy parameter $\theta_0$;
**for** $k = 1, 2, \ldots$ **do**
    Generate $\{(S_t^k, A_t^k, r(S_t^k, A_t^k))\}_{t=1}^n$ on $\pi_\theta$;
    Update policy parameters $\theta_{k+1} \leftarrow \theta_k + \eta_i \widehat{\nabla} \mathbb{E}_{\pi_{\theta_k}} \left[ u^{\text{trajectory}}(G; \alpha, \beta) \right]$.
**end for**

---

**Theorem 1.** *Consider an update rule such that $\theta_{k+1} \leftarrow \theta_k + \eta_i \widehat{\nabla} \mathbb{E}_{\pi_{\theta_k}} \left[ u^{\text{trajectory}}(G; \alpha, \beta) \right]$, where the learning rates $\eta_k$ are non-negative and satisfy $\sum_{k=0}^{\infty} \eta_k = \infty$ and $\sum_{k=0}^{\infty} \eta_k^2 < \infty$. Suppose that (a) episode always finishes in finite horizon $n$; (b) the policy $\pi_\theta$ has always bounded first and second partial derivatives. Then, $\lim_{i \to \infty} \nabla_\theta \mathbb{E}_{\pi_{\theta_i}} [\alpha G - \frac{1}{2} \beta G^2] = 0$ almost surely.*

It is expected that non-asymptotic results can be derived by restricting the policy class and the optimization algorithm as Agarwal et al. (2020) and Zhang et al. (2021a), but this is not the scope of this paper, which aims to provide a general framework.

**AC-based trajectory EQUMRL.** Another implementation of the EQUMRL is to apply the AC algorithm (Williams & Peng, 1991; Mnih et al., 2016). For an episode $k$ with the length $n$, following Prashanth & Ghavamzadeh (2013; 2016), we train the policy by a gradient defined as

$$\nabla_\theta \log \pi_\theta(S_t^k, A_t^k) \left\{ \left( \alpha \widetilde{G}_{t:t+n-1}^k - \frac{1}{2} \beta \widetilde{G}_{t:t+n-1}^{k,2} \right) - \left( \alpha M_{\hat{\omega}_k^{(1)}}^{(1)}(S_t^k) - \frac{1}{2} \beta M_{\hat{\omega}_k^{(2)}}^{(2)}(S_t^k) \right) \right\},$$

where $\widetilde{G}_{t:t+n-1}^k = \widehat{G}_{t:t+n-1}^k + \gamma^n M_{\hat{\omega}_k^{(1)}}^{(1)}(S_{t+n}^k)$, $\widetilde{G}_{t:t+n-1}^{k,2} = \left( \widehat{G}_{t:t+n-1}^k \right)^2 + 2\gamma^n \widehat{G}_{t:t+n-1}^k M_{\hat{\omega}_k^{(1)}}^{(1)}(S_{t+n}^k) + \gamma^{2n} M_{\hat{\omega}_k^{(2)}}^{(2)}(S_{t+n}^k)$, and $M_{\hat{\omega}_k^{(1)}}^{(1)}(S_t^k)$ and $M_{\hat{\omega}_k^{(2)}}^{(2)}(S_t^k)$ are models of $\mathbb{E}[G_{t+1:\infty}]$ and $\mathbb{E}[G_{t+1:\infty}^2]$ with parameters $\hat{\omega}_k^{(1)}$ and $\hat{\omega}_k^{(2)}$. For more details, see Prashanth & Ghavamzadeh (2013; 2016).

**Remark 1** (Existing approaches). For the double sampling issue, Tamar et al. (2012) and Prashanth & Ghavamzadeh (2013; 2016) proposed multi-time-scale stochastic optimization. Their approaches are known to be sensitive to the choice of step-size schedules, which are not easy to control (Xie et al., 2018). Xie et al. (2018) proposed using the Legendre-Fenchel dual transformation with coordinate descent algorithm. First, based on Lagrangian relaxation, Xie et al. (2018) set an objective function as $\max_{\theta \in \Theta} \mathbb{E}_{\pi_\theta}[G] - \delta \left( \mathbb{V}_{\pi_\theta}(G) - \eta \right)$. Then, Xie et al. (2018) transformed the objective function as $\max_{\theta \in \Theta, y \in \mathbb{R}} 2y \left( \mathbb{E}_{\pi_\theta}[G] + \frac{1}{2\delta} \right) - y^2 - \mathbb{E}_{\pi_\theta} \left[ G^2 \right]$ and trained an agent by solving the optimization problem via a coordinate descent algorithm. However, this approach does not reflect the constraint $\eta$ because the constraint condition $\eta$ vanishes from the objective function. This problem is caused by their objective function based on the penalty function $g(x) = x$: $\mathbb{E}_{\pi_\theta}[G] - \delta \left( \mathbb{V}_{\pi_\theta}(G) - \eta \right)$, where the first derivative does not include $\eta$. To avoid this problem, we need an iterative algorithm to decide an optimal $\delta$ or change $g(x)$ from $x$ but it is not obvious how to incorporate them into the approach.

**Remark 2** (Difference from existing MV approaches). Readers may assert that EQUMRL simply omits $\left( \mathbb{E}_{\pi_\theta}[G] \right)^2$ from existing MV-controlled RL methods, which usually includes the explicit variance term in the objective function, and is the essentially the same. However, there are significant differences; one of the main findings of this paper is our formulation of a simpler RL problem to obtain an MV-efficient policy. Existing MV-controlled RL methods suffer from computational difficulties caused by the double sampling issue. However, we can obtain MV-efficient policy without going through the difficult problem. In addition, EQUMRL shows better performance in experiments even from the viewpoint of the constrained problem because it is difficult to choose parameters to avoid the double sampling issue in existing approaches. Thus, the EQUMRL has advantage in avoiding solving more difficult constrained problems for considering MV trade-off.

### 4.3 INTERPRETATIONS OF EQUMRL WITH GRADEINT ESTIMATION

We can interpret EQUMRL as an approach for (i) a targeting optimization problem to achieve an expected cumulative reward $\zeta$, (ii) an expected cumulative reward maximization with regularization, and (iii) expected utility maximization via Taylor approximation.

First, we can also interpret EQUMRL as mean squared error (MSE) minimization between a cumulative reward $R$ and a target return $\zeta$; that is,

$$\arg\min_{\theta \in \Theta} J(\theta; \zeta) = \arg\min_{\theta \in \Theta} \mathbb{E}_{\pi_\theta} \left[ (\zeta - G)^2 \right] \tag{4}$$

We can decompose the MSE into the bias and variance as

$$\mathbb{E}_{\pi_\theta} \left[ (\zeta - G)^2 \right] = \underbrace{(\zeta - \mathbb{E}_{\pi_\theta}[G])^2}_{\text{Bias}} + \underbrace{2\mathbb{E}_{\pi_\theta} \left[ (\zeta - \mathbb{E}_{\pi_\theta}[G]) (\mathbb{E}_{\pi_\theta}[G] - G) \right]}_{0} + \underbrace{\mathbb{E}_{\pi_\theta} \left[ (\mathbb{E}_{\pi_\theta}[G] - G)^2 \right]}_{\text{Variance}}$$

$$= \zeta^2 - 2\zeta\mathbb{E}_{\pi_\theta}[G] + (\mathbb{E}_{\pi_\theta}[G])^2 + \mathbb{E}_{\pi_\theta} \left[ (\mathbb{E}_{\pi_\theta}[G] - G)^2 \right].$$

Thus, the MSE minimization (4) is equivalent to EQUMRL (3), where $\zeta = \frac{\alpha}{\beta}$. The above equation provides an important implication for the setting of $\zeta$. If we know the reward is shifted by $x$, we only have to adjust $\zeta$ to $\zeta + x$. This is because $\zeta$ only affects the bias term in the above equation. The equation also provides another insight. If our assumption $\max_{\pi_\theta} \mathbb{E}_{\pi_\theta}[G] \leq \zeta$ is violated, $\mathbb{E}_{\pi_\theta}[G]$ will not be maximized with a fixed variance and will be biased towards $\zeta$; that is, EQUMRL cannot find the MV-efficient policies. In applications, we can confirm whether the optimization works by checking whether average value of the empirically realized cumulative rewards is less than $\zeta$.

Second, we can regard the quadratic utility function as an expected cumulative reward maximization with a regularization term defined as $\mathbb{E}[R^2]$; that is, minimization of the risk $\mathcal{R}(\pi_\theta)$:

$$\mathcal{R}(\theta) = \underbrace{-\mathbb{E}_{\pi_\theta}[G]}_{\text{Risk of expected cumulative reward maximization}} + \underbrace{\psi\mathbb{E}_{\pi_\theta}[G^2]}_{\text{Regularization term}}$$

where $\psi > 0$ is a regulation parameter and $\psi = \frac{\beta}{2\alpha} = \frac{1}{2\zeta}$. As $\psi \to 0$ ($\zeta \to \infty$), $\mathcal{R}(\theta) \to -\mathbb{E}_{\pi_\theta}[G]$.

Third, the quadratic utility function is the quadratic Taylor approximation of a smooth utility function $u(G)$ because for $G_0 \in \mathbb{R}$, we can expand it as $u(G) \approx u(G_0) + U'(G_0)(G - G_0) + U''(G_0)(G - G_0)^2 + \cdots$; that is, quadratic utility is an approximation of various risk-averse utility functions. This property also supports the use of the quadratic utility function in practice (Kroll et al., 1984).

It also should be noted that the EQUMRL is closely related to the fields of economics and finance, where the ultimate goal is to maximize the utility of an agent, which is also referred to as an *investor*. The quadratic utility function is a standard risk-averse utility function often assumed in financial theory Luenberger et al. (1997) to justify an MV portfolio; that is, an MV portfolio maximizes an investor's utility function if the utility function is quadratic (see Appendix A). Therefore, our approach can be interpreted as a method that directly achieves the original goal.

**Remark 3** (Specification of utility function (hyper-parameter selection)). Next, we discuss how to decide the parameters $\alpha$ and $\beta$, which are equivalent to $\zeta$, $\xi$, and $\psi$. The meanings are equivalent to constrained conditions of MVRL; that is, we predetermine these hyperparameters depending on our attitude toward risk. For instance, we propose the following three directions for the parameter choice. First, we can determine $\frac{\beta}{2\alpha} = \psi$ based on economic theory or market research (Ziemba et al., 1974; Kallberg et al., 1983) (Appendix A). Luenberger et al. (1997) proposed some questionnaires to investors for the specification. Second, we set $\zeta = \frac{1}{2\psi}$ as the targeted reward that investors aim to achieve. Third, through cross-validation, we can optimize the regularization parameter $\psi$ to maximize some criteria, such as the Sharpe ratio (Sharpe, 1966). However, we note that in time-series related tasks, we cannot use standard cross-validation owing to dependency. Therefore, in our experiments with a real-world financial dataset, we show various results under different parameters.

**Remark 4** (From MV-efficient RL to MV-controlled RL). We can also apply the MV-efficient RL method as the MV-controlled RL method. The parameters of the expected quadratic utility function correspond to the variance that we want to achieve. For example, the larger $\zeta$ is in (4), the larger the variance will be. Although we do not know the explicit correspondence between the parameter of the expected quadratic utility function and the variance, it is possible to control the variance by choosing an appropriate policy from among those learned under several parameters. In Figure 2 in Section 6, we show the mean and variance of several policies trained with several parameters. These results are measured with test data, but by outputting multiple candidates with the training data (ex. cross-validation), we can choose a policy with the desired variance.

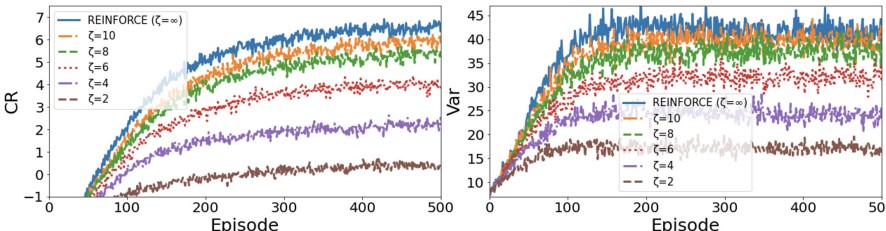

Figure 1: The CRs and Vars in the training process of the experiment using the synthetic dataset.

## 5 EQUMRL UNDER PER-STEP VARIANCE PERSPECTIVE

Another direction of MVRL is to consider the trade-off between the per-step mean $\mathbb{E}_{\pi_\theta}[r(S_t, A_t)]$ and per-step variance $\mathbb{V}_{\pi_\theta}(r(S_t, A_t))$, $\mathbb{E}_{\pi_\theta}[r(S_t, A_t)] \overset{\text{trade-off}}{\iff} \mathbb{V}_{\pi_\theta}(r(S_t, A_t))$. As well as the trajectory perspective, Bisi et al. (2020) and Zhang et al. (2021b) proposed training a policy to maximize the penalized objective function, $\max_{\theta \in \Theta} \sum_{t=1}^{T} \gamma^{t-1}\Big(\mathbb{E}_{\pi_\theta}[r(S_t, A_t)] - \lambda \mathbb{V}_{\pi_\theta}(r(S_t, A_t))\Big)$. We call an algorithm with this formulation a per-step MV-controlled RL. See Appendix B for more details.

We define a per-step MV-efficient policy as a policy $\pi_\theta$ such that there is no other policy with $\theta' \in \Theta$, where, for each $t$, $\mathbb{E}_{\pi_\theta}[r(S_t, A_t)] \leq \mathbb{E}_{\pi_{\theta'}}[r(S_t, A_t)]$ and $\mathbb{V}_{\pi_\theta}[r(S_t, A_t)] > \mathbb{V}_{\pi_{\theta'}}[r(S_t, A_t)]$, or $\mathbb{V}_{\pi_\theta}[r(S_t, A_t)] \geq \mathbb{V}_{\pi_{\theta'}}[r(S_t, A_t)]$ and $\mathbb{E}_{\pi_\theta}[r(S_t, A_t)] < \mathbb{E}_{\pi_{\theta'}}[r(S_t, A_t)]$. For constants $\alpha, \beta > 0$, we define the quadratic utility function for per-step reward setting as $u^{\text{per-step}}(r(S_t, A_t); \alpha, \beta) = \alpha r(S_t, A_t) - \beta r^2(S_t, A_t)$. For an finite horizon case with $\gamma = 1$, we consider maximizing the expected cumulative quadratic utility function defined as $\mathbb{E}_{\pi_\theta}\left[\sum_{t=1}^{T} u^{\text{per-step}}(r(S_t, A_t); \alpha, \beta)\right] = \alpha\mathbb{E}_{\pi_\theta}\left[\sum_{t=1}^{T} r(S_t, A_t)\right] - \beta\mathbb{E}_{\pi_\theta}\left[\sum_{t=1}^{T} r^2(S_t, A_t)\right]$. When applying REINFORCE-based algorithm to train an agent, the sample approximation of the gradient is given as $\widehat{\nabla}\mathbb{E}_{\pi_\theta}\left[\sum_{t=1}^{T} u^{\text{per-step}}(r(S_t^k, A_t^k); \alpha, \beta)\right] = \sum_{t=1}^{T}\{\alpha r(S_t^k, A_t^k) - \beta(r(S_t^k, A_t^k))^2\}\nabla_\theta \log \pi_\theta(S_t^k, A_t^k)$. Appendix B contains the performance using a synthetic dataset.

## 6 EXPERIMENTS

This section investigates the empirical performance of the proposed EQUMRL in trajectory variance setting using synthetic and real-world financial datasets. We train the policy by REINFORCE-based algorithm. We conduct two experiments. In the first experiments, following Tamar et al. (2012; 2014), and Xie et al. (2018), we conduct portfolio management experiments with synthetic datasets. In the third experiment, we conduct a portfolio management experiment with a dataset of Fama & French (1992), a standard benchmark in finance. In Appendix C, following Tamar et al. (2012; 2014), and Xie et al. (2018), we also conduct American-style option experiments with synthetic datasets. We implemented algorithms following the Pytorch example[1]. For algorithms using neural networks, we use a three layer perceptron, where the numbers of the units in two hidden layers are the same as that of the input node, and that of the output node is 2. We note that in all results, naively maximizing the reward or minimizing the variance do not ensure a better algorithm; we evaluate an algorithm based on how it controls the MV trade-off. We denote the hyperparameter of the EQUMRL by $\zeta$, which has the same meaning as $(\alpha, \beta)$ and $\psi$. For all experiments, we adopt episodic MDPs; that is, $\gamma = 1$.

### 6.1 PORTFOLIO MANAGEMENT WITH A SYNTHETIC DATASET

Following Tamar et al. (2012) and Xie et al. (2018), we consider a portfolio composed of two asset types: a liquid asset with a fixed interest rate $r_l$, and a non-liquid asset with a time-dependent interest rate taking either $r_{\text{nl}}^{\text{low}}$ or $r_{\text{nl}}^{\text{high}}$, and the transition follows a switching probability $p_{\text{switch}}$. An investor can sell the liquid asset at every time step $t = 1, 2, \dots, T$ but the non-liquid asset can only be sold after the maturity $W$ periods. This means that when holding 1 liquid asset, we obtain $r_l$ per period; when holding 1 non-liquid asset at the $t$-th period, we obtain $r_{\text{nl}}^{\text{low}}$ or $r_{\text{nl}}^{\text{high}}$ at the $t + W$-th period.

---

[1]https://github.com/pytorch/examples/tree/master/reinforcement_learning

Table 1: The results of the portfolio management with a synthetic data. The upper table shows the CRs and Vars over $100,000$ trials. The lower table shows the MSEs for $\zeta$ over $100,000$ trials.

| | REINFORCE | EQUM | | | Tamar | | Xie | |
|---|---|---|---|---|---|---|---|---|
| | ($\zeta = \infty$) | $\zeta = 10$ | $\zeta = 6$ | $\zeta = 4$ | $\mathbb{V} = 80$ | $\mathbb{V} = 50$ | $\lambda = 100$ | $\lambda = 10$ |
| CR | 6.729 | 6.394 | 4.106 | 2.189 | 6.709 | 2.851 | 0.316 | 0.333 |
| Var | 32.551 | 31.210 | 24.424 | 18.518 | 32.586 | 21.573 | 15.883 | 15.992 |

| | REINFORCE | EQUM | | | | |
|---|---|---|---|---|---|---|
| Target Value | ($\zeta = \infty$) | $\zeta = 10$ | $\zeta = 8$ | $\zeta = 6$ | $\zeta = 4$ | $\zeta = 2$ |
| MSE from $\zeta = 10$ | **51.669** | 53.399 | 55.890 | 65.307 | 83.061 | 105.492 |
| $\zeta = 8$ | **42.586** | 42.975 | 43.375 | 45.730 | 55.816 | 71.480 |
| $\zeta = 6$ | 41.503 | 40.551 | 38.860 | **34.154** | 36.570 | 45.467 |
| $\zeta = 4$ | 48.420 | 46.127 | 42.345 | 30.577 | **25.324** | 27.455 |
| $\zeta = 2$ | 63.337 | 59.703 | 53.830 | 35.000 | 22.078 | **17.442** |

Besides, the non-liquid asset has a risk of not being paid with a probability $p_{\text{risk}}$; that is, if the non-liquid asset defaulted during the $W$ periods, we could not obtain any rewards by having the asset. An investor can change the portfolio by investing a fixed fraction $w$ of the total capital $M$ in the non-liquid asset at each time step. A typical investment strategy is to construct a portfolio using both liquid and non-liquid assets for decreasing the variance. Following Xie et al. (2018), we set $r_l = 1.001$, $r_{\text{nl}}^{\text{low}} = 1.1$, $r_{\text{nl}}^{\text{high}} = 2$, $p_{\text{switch}} = 0.1$, $p_{\text{risk}} = 0.05$, $W = 4$, $w = 0.2$, and $M = 1$. As a performance metric, we use the average cumulative reward (CR) and its variance (Var) when investing for 50 periods. We compare the EQUMRL with the REINFORCE, MV-controlled methods proposed by Tamar et al. (2012) (Tamar), and Xie et al. (2018) (Xie). We denote the variance constraint of Tamar et al. (2012) as $\mathbb{V}$ar and Lagrange multiplier of Xie et al. (2018) as $\lambda$. For training Tamar, Xie, and EQUMRL, we set the Adam optimizer with learning rate $0.01$ and weight decay parameter $0.1$. For each algorithm, we report performances under various hyperparameters as much as possible.

First, we show CRs and Vars of the EQUMRL during the training process in Figure 1, where we conduct 100 trials on the test environment to compute CRs and Vars for each episode. Here, we also show the performance of REINFORCE, which corresponds to the EQUMRL with $\zeta = \alpha/\beta = \infty$. As Figure 1 shows, the EQUMRL trains MV-efficient policies well depending on the parameter $\zeta$. Next, we compare the EQUMRL on the test environment with the REINFORCE, Tamar, and Xie. We conduct $100,000$ trials on the test environment to compute CRs and Vars.

In Figure 2, we plot performances under several hyperparameters, where the horizontal axis denotes the Var, and the vertical axis denotes the CR. Trained agents with a higher CR and lower Var are Pareto efficient. As the result shows, the EQUMRL returns more efficient portfolios than the others in almost all cases. We conjecture that this is because while the EQUMRL is an end-to-end optimization for obtaining an efficient agent, the other methods consist of several steps for solving the constrained optimization, where those multiple steps can be sources of the suboptimal result. We show CRs and Vars of some of their results in the upper table of Table 1. The MSEs between $\zeta$ and CR are also shown in the lower table of Table 1, where we can confirm that the EQUMRL succeeded in minimizing the MSEs.

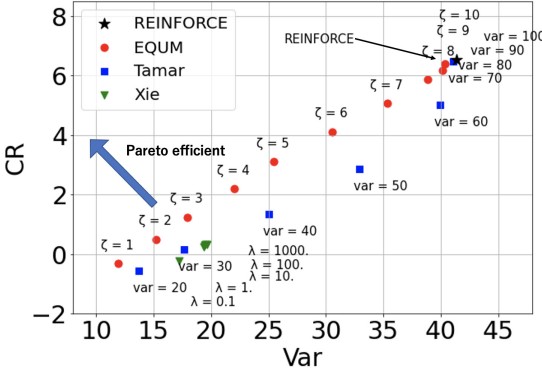

Figure 2: MV efficiency of the portfolio management experiment. Higher CRs and lower Vars methods are MV Pareto efficient.

## 6.2 PORTFOLIO MANAGEMENT WITH A REAL-WORLD DATASET

We use standard benchmarks called Fama & French (FF) datasets[2] to ensure the reproducibility (Fama & French, 1992). Among FF datasets, we use the FF25, FF48 and FF100 datasets, where the FF25 and FF 100 dataset includes 25 and 100 assets formed based on size and book-to-market ratio; the FF48 dataset contains 48 assets representing different industrial sectors. We use all datasets covering

---

[2] https://mba.tuck.dartmouth.edu/pages/faculty/ken.french/data_library.html

Table 2: The performance of each portfolio model during the out-of-sample period (from July 2000 to June 2020) for FF25 dataset (upper table) , FF48 (middle table), and FF100 (lower table). For each dataset, the best performance is highlighted in **bold**.

| Method | EW | MV | EGO | BLD | Tamar | | | Xie | | | EQUM | | |
|---|---|---|---|---|---|---|---|---|---|---|---|---|---|
| | | | | | $\mathbb{V}=15$ | $\mathbb{V}=30$ | $\mathbb{V}=60$ | $\lambda=10$ | $\lambda=100$ | $\lambda=1000$ | $\zeta=0.5$ | $\zeta=0.75$ | $\zeta=1.5$ |
| | | | | | | | FF25 | | | | | | |
| CR↑ | 0.80 | 0.09 | 0.81 | 0.84 | 1.15 | 1.01 | 0.80 | 0.90 | 1.15 | 1.15 | **1.53** | 1.25 | 1.11 |
| Var↓ | 28.62 | 53.57 | 30.65 | 22.16 | 18.53 | 15.04 | 25.29 | 25.96 | 25.77 | 25.77 | 24.27 | 15.02 | **11.77** |
| R/R↑ | 0.52 | 0.04 | 0.51 | 0.62 | 0.93 | 0.90 | 0.55 | 0.61 | 0.79 | 0.79 | 1.07 | 1.12 | **1.13** |
| MaxDD↓ | 0.54 | 0.75 | 0.58 | 0.52 | 0.35 | 0.35 | 0.56 | 0.54 | 0.51 | 0.51 | 0.36 | 0.31 | **0.27** |
| | | | | | | | FF48 | | | | | | |
| CR↑ | 0.81 | 0.11 | 0.97 | 0.75 | 0.68 | 0.38 | 0.82 | 0.50 | 1.08 | 1.01 | **1.60** | 1.05 | 0.91 |
| Var↓ | 22.91 | 77.02 | 31.91 | 15.98 | 40.69 | 16.00 | 18.04 | 27.12 | 23.77 | 26.31 | 31.97 | 18.69 | **10.34** |
| R/R↑ | 0.59 | 0.04 | 0.60 | 0.65 | 0.37 | 0.33 | 0.67 | 0.33 | 0.76 | 0.68 | **0.98** | 0.84 | 0.98 |
| MaxDD↓ | 0.30 | 0.48 | 0.31 | 0.25 | 0.32 | 0.20 | 0.21 | 0.27 | 0.25 | 0.26 | 0.29 | 0.21 | **0.16** |
| | | | | | | | FF100 | | | | | | |
| CR↑ | 0.81 | 0.11 | 0.81 | 0.85 | 1.13 | 0.79 | 0.67 | 0.57 | 0.98 | 1.24 | 0.95 | 0.95 | **1.43** |
| Var↓ | 29.36 | 57.97 | 32.35 | 21.83 | 25.50 | 20.34 | 16.50 | 87.92 | 28.78 | 41.06 | 15.50 | **14.26** | 33.09 |
| R/R↑ | 0.52 | 0.05 | 0.49 | 0.63 | 0.77 | 0.61 | 0.57 | 0.21 | 0.63 | 0.67 | 0.83 | **0.87** | 0.86 |
| MaxDD↓ | 0.33 | 0.46 | 0.34 | 0.27 | 0.25 | 0.23 | 0.25 | 0.51 | 0.32 | 0.31 | **0.19** | 0.26 | 0.31 |

monthly data from July 1980 to June 2040. Consider an episodic MDP. Let the action space be the set of $m$ assets, and $y_{a,t}$ and $w_{a,t}$ be the return and the portfolio weight of an asset $a$ at time $t$. The reward (portfolio return) at time $1 \le t \le T$ is defined as $y_t = \sum_{a=1}^m y_{a,t} w_{a,t} - \Lambda \sum_{a=1}^m |w_{a,t} - w_{a,t-1}|$, where $\Lambda = 0.001$ is the penalty of the portfolio weight turnover(change of the action). On the $t$-th period, the agent observes a state $S_t = ((y_{a,t-1}, ..., y_{a,t-12}, w_{a,t-1})_{a \in \mathcal{A}}, \sum_{s=0}^{t-1} y_s)$, and decides a portfolio weight $(w_{a,t})_{a \in \mathcal{A}}$ as $w_{a,t} = \pi(a, s_t)$. Between periods $t$ and $t+1$, the agent has an asset $a$ with the ratio $w_{a,t}$.See Section 4.1 for the reason why we include $sum_{s=0}^{t-1} y_s$ in the state.

We use the following portfolio models: an equally-weighted portfolio (EQ, DeMiguel et al., 2009); a mean-variance portfolio (MV, Markowitz, 1952); a Kelly growth optimal portfolio Shen et al. (2019); Portfolio blending via Thompson sampling (BLD, Shen & Wang, 2016). We also compare our proposed method with the methods proposed by Tamar et al. (2012) and Xie et al. (2018). Denote a method proposed by Tamar et al. (2012) by Tamar, and choose the parameter var from $\{15, 30, 60\}$. Denote a method proposed by Xie et al. (2018) by Xie, and choose the parameter $\lambda$ from $\{10, 100, 1000\}$. Denote the REINFORCE-based trajectory EQUMRL by EQUMRL, and choose the parameter $\zeta$ from $\{0.5, 0.75, 1.5\}$.

We apply the following standard measures in finance for evaluation (Brandt, 2010). The cumulative reward (CR), annualized risk as the standard deviation of return (RISK) and risk-adjusted return (R/R) are defined as follows: $\mathrm{CR} = 1/T \sum_{t=1}^T y_t$, $\mathbb{V} = 1/T \sum_{t=1}^T (y_t - \mathrm{CR})^2$, and $\mathrm{R/R} = \sqrt{12} \times \mathrm{CR}/\sqrt{\mathrm{Var}}$. R/R is the most important measure for a portfolio strategy and is often referred to as the Sharpe Ratio (Sharpe, 1966). We also evaluate the maximum draw-down (MaxDD), which is another widely used risk measure (Magdon-Ismail & Atiya, 2004). In particular, MaxDD is the largest drop from a peak defined as $\mathrm{MaxDD} = \min_{t \in [1,T]} \left(0, \frac{W_t}{\max_{\tau \in [1,t]} W_\tau} - 1\right)$, where $W_k$ is the cumulative return of the portfolio until time $k$; that is, $W_t = \prod_{t'=1}^t (1 + y_{t'})$.

Table 2 reports the performances of the portfolios. In almost all cases, the EQUMRL achieves the highest R/R and the lowest MaxDD. Therefore, we can confirm that the EQUMRL has a high R/R, and avoids a large drawdown. The real objective (minimizing variance with a penalty on return targeting) for Tamar, MVP, and EQUMRL is shown in Appendix D. Except for FF48's MVP, the objective itself is smaller than that of EQUMRL. Since the values of the objective are proportional to R/R, we can empirically confirm that the better optimization, the better performance.

## 7    CONCLUSION

In this paper, we proposed EQUMRL for MV-efficient RL. Compared with the conventional MVRL methods, EQUMRL is computationally friendly. The proposed EQUMRL also includes various interpretations, such as targeting optimization and regularization, which expands the scope of applications of the method. We investigated the effectiveness of the EQUMRL compared with the standard RL and existing MVRL methods through experiments using synthetic and real-world datasets.

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

# A PRELIMINARIES OF ECONOMIC AND FINANCIAL THEORY

## A.1 UTILITY THEORY

Utility theory is the foundation of the choice theory under uncertainty including economics and financial theory. A utility function $u(\cdot)$ measures agent's relative preference for different levels of total wealth $W$. According to Morgenstern & Von Neumann (1953), a rational agent makes an investment decision to maximize the expected utility of wealth among a set of competing feasible investment alternatives. For simplicity, the following assumptions are often made for the utility function used in economics and finance. First, the utility function is assumed to be at least twice continuous differentiable. The first derivative of the utility function (the marginal utility of wealth) is always positive, i.e., $U'(W) > 0$, because of the assumption of non-satiation. The second assumption concerns risk attitude of the agents called "risk averse". When we assume that an agent is risk averse, the utility function is described as a curve that increases monotonically and is concave. The most often used utility function of a risk-averse agent is the quadratic utility function as follows:

$$u(W; \alpha, \beta) = \alpha W - \beta \frac{1}{2} W^2 \tag{5}$$

where $\alpha \geq 0, \beta > 0$. Taking the expected value of the quadratic utility function in (5) yields:

$$E[u(W; \alpha, \beta)] = \alpha \mathbb{E}[W] - \frac{1}{2} \beta \mathbb{E}[W^2] \tag{6}$$

Substituting $\mathbb{E}[W^2] = \mathbb{V}[W] + \mathbb{E}[W]^2$ into (5) gives

$$\mathbb{E}[u(W; \alpha, \beta)] = \alpha \mathbb{E}[W] - \frac{1}{2} \beta (\mathbb{V}[W] + \mathbb{E}[W]^2) \tag{7}$$

Equation (7) shows that expected quadratic utility can be described in terms of mean $E[W]$ and variance $\mathbb{V}[W]$ of wealth. Therefore, the assumption of a quadratic utility function is crucial to the mean-variance analysis.

**Remark 5** (Approximation by quadratic utility function). Readers may be interested in how the quadratic utility function approximates other risk averse utility functions. Kroll et al. (1984) empirically answered this question by comparing MV portfolio (maximizer of expected quadratic utility function) and maximizers of other utility functions. In their study, maximizers of other utility functions also almost located in MV Pareto efficient frontier; that is, expected quadratic utility function approximates other risk averse utility function well.

**Remark 6** (Non-vNM utility functions). Unlike MV trade-off, utility functions maximized by some recently proposed new risk criteria, such as VaR and Prospect theory, do not belong to traditional vNM utility function.

## A.2 MARKOWITZ'S PORTFOLIO

Considering the mean-variance trade-off in a portfolio and economic activity is an essential task in economics as Tamar et al. (2012) and Xie et al. (2018) pointed out. The mean-variance trade-off is justified by assuming either quadratic utility function to the economic agent or multivariate normal distribution to the financial asset returns (Borch, 1969; Baron, 1977; Luenberger et al., 1997). This means that if either the agent follows the quadratic utility function or asset return follows the normal distribution, the agent's expected utility function is maximized by maximizing the expected reward and minimizing the variance. Therefore, the goal of Markowitz's portfolio is not only to construct the portfolio itself but also to maximize the expected utility function of the agent. See Figure 3.

Markowitz (1952) proposed the following steps for constructing a MV controlled portfolio (Also see Markowitz (1959), page 288, and Luenberger et al. (1997)):

- Constructing portfolios minimizing the variance under several reward constraint;
- Among the portfolios constructed in the first step, the economic agent chooses a portfolio maximizing the utility function.

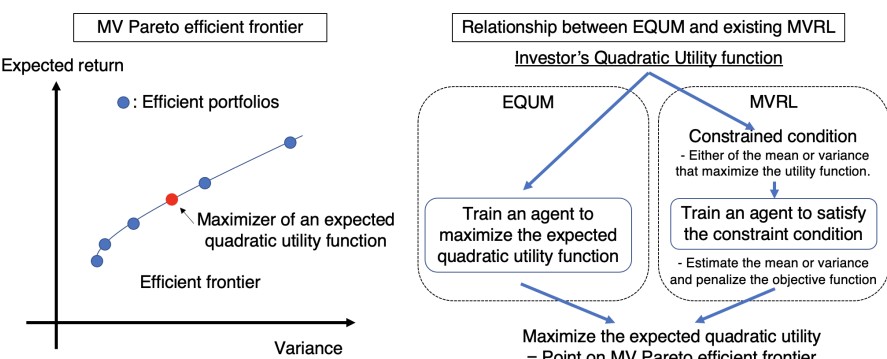

Figure 3: The concept of MV Preto efficient frontier and difference between EQUMRL and existing MVRL approaches.

Conventional financial methods often adopt this two-step approach because directly predicting the reward and variance to maximize the expected utility function is difficult; therefore, first gathering information based on analyses of an economist, then we construct the portfolio using the information and provide the set of the portfolios to an economic agent. However, owing to the recent development of machine learning, we can directly represent the complicated economic dynamics using flexible models, such as deep neural networks. In addition, as Tamar et al. (2012) and Xie et al. (2018) reported, when constructing the mean-variance portfolio in RL, we suffer from the double sampling issue. Therefore, this paper aims to achieve the original goal of the mean-variance approach; that is, the expected utility maximization. Note that this idea is not restricted to financial applications but can be applied to applications where the agent utility can be represented only by the mean and variance.

### A.3   Markowitz's portfolio and capital asset pricing model

Markowitz's portfolio is known as the mean-variance portfolio (Markowitz, 1952; Markowitz & Todd, 2000). Constructing the mean-variance portfolio is motivated by the agent's expected utility maximization. When the utility function is given as the quadratic utility function, or the financial asset returns follow the multivariate normal distribution, a portfolio maximizing the agent's expected utility function is given as a portfolio with minimum variance under a certain standard expected reward.

The Capital Asset Pricing Model (CAPM) theory is a concept which is closely related to Markowitz's portfolio (Sharpe, 1964; Lintner, 1965; Mossin, 1966). This theory theoretically explains the expected return of investors when the investor invests in a financial asset; that is, it derives the optimal price of the financial asset. To derive this theory, as well as Markowitz's portfolio, we assume the quadratic utility function to the investors or the multivariate normal distribution to the financial assets.

Merton (1969) extended the static portfolio selection problem to a dynamic case. Fishburn & Burr Porter (1976) studied the sensitivity of the portfolio proportion when the safe and risky asset distributions change under the quadratic utility function. Thus, there are various studies investigating relationship between the utility function and risk-averse optimization (Tobin, 1958; Kroll et al., 1984; Bulmuş & Özekici, 2014; Bodnar et al., 2015a;b).

### A.4   MV portfolio and MVRL

Traditional portfolio theory have attempted to maximize the expected quadratic utility function by providing MV portfolios. This is because MV portfolio is easier to interpret than EQUMRL, and we can obtain a solution by quadratic programming. One of the main goals of MVRL methods is also to construct MV portfolio under a dynamic environment (Tamar et al., 2012). However, we conjecture that there are three significant differences between them. First, unlike static problem, MVRL suffers computational difficulties. Second, while static problem solves quadratic programming given the expected reward and variance, MVRL methods simultaneously estimate these values and solve the constrained problem. Third, while static MV portfolios gives us an exact solution of the constrained problem, MVRL often relaxes the constrained problem by penalized method, which

causes approximation errors. In particular, for the second and third points, the difference in how to handle the estimators of expected reward and variance is essential.

### A.5 EMPIRICAL STUDIES ON THE UTILITY FUNCTIONS

The standard financial theory is built on the assumption that the economic agent has the quadratic utility function. For supporting this theory, there are several empirical studies to estimate the parameters of the quadratic utility function. Markowitz & Todd (2000) discussed how the quadartic utility function approximates the other risk-averse utility functions. Ziemba et al. (1974) investigated the change of the portfolio proportion when the parameter of the quadratic utility function changes using the Canadian financial dataset. Recently, Bodnar et al. (2018) investigate the risk parameter ($\alpha$ and $\beta$ in our formulation of the quadratic utility function) using the markets indexes in the world. They found that the utility function parameter depends on the market data model.

### A.6 CRITICISM

For the simple form of the quadratic utility function, the financial models based on the utility are widely accepted in practice. However, there is also criticism that the simple form cannot capture the real-world complicated utility function. For instance, Kallberg et al. (1983) criticized the use of the quadratic utility function and proposed using a utility function, including higher moments. This study also provided empirical studies using U.S. financial dataset for investigating the properties of the alternative utility functions. However, to the best of our knowledge, financial practitioners still prefer financial models based on the quadratic utility function. We consider this is because the simple form gains the interpretability of the financial models.

### A.7 ECONOMICS AND FINANCE

To mathematically describe an attitude toward risk, economics and finance developed expected utility theory, which assumes the Bernoulli utility function $u(R)$ on an agent. In the expected utility theory, an agent acts to maximize the von Neumann-Morgenstern (vNM) utility function $U(F(r)) = \int u(r)dF(r)$, where $F(r)$ is a distribution of $R$ (Mas-Colell et al., 1995). We can relate the utility function form to agents with three different risk preferences: the utility function $u(R)$ is concave for risk averse agents; linear for risk neutral agents; and convex risk seeking agents. For instance, an agent with $u(R) = R$ corresponds to a risk neutral agent attempting to maximize their expected cumulative reward in a standard RL problem. For more detailed explanation, see Appendix A or standard textbooks of economics and finance, such as Mas-Colell et al. (1995) and Luenberger et al. (1997).

To make the Bernoulli utility function more meaningful, we assume that it is increasing function with regard to $R$; that is, $R \leq \frac{\alpha}{\beta}$ for all possible $R$. Even without the assumption, for a given pair of $(\alpha, \beta)$, a optimal policy maximizing the expected utility does not change; that is, the assumption is only related to interpretation of the quadratic utility function and does not affect the optimization.

On the other hand, the constraint condition is determined by an investor to maximize its expected utility (Luenberger et al., 1997). Finally, in theories of economics and finance, investors can maximize their utility by choosing a portfolio from MV portfolios. In addition, when $R \leq \frac{\alpha}{\beta}$ for all possible value of $R$, a policy maximizing the expected utility function is also located on MV Pareto efficient frontier.

## B PER-STEP VARIANCE PERSPECTIVE

Bisi et al. (2020) defines the per-step reward random variable $R$, a discrete random variable taking the values in the image of $r$, by defining its probability mass function as $p(R = x) = \sum_{s,a} d_{\pi_\theta}(s,a)\mathbb{1}[r(s,a) = x]$, where $\mathbb{1}$ is the indicator function and $d_{\pi_\theta}$ is the normalized discounted state-action distribution, $d_{\pi_\theta} = (1 - \gamma)\sum_{t=0}^{\infty}\gamma^t\Pr_{\mu_0,\pi_\theta,p}(S_t = s, A_t = a)$. Here, it is known that for $\gamma < 1$, $\mathbb{E}_{\pi_\theta}[G] = \frac{1}{1-\gamma}\sum_{s,a} d_{\pi_\theta}(s,a)r(s,a)$. It follows that $\mathbb{E}_{\pi_\theta}[R] = (1 - \gamma)J(\pi_\theta)$. Bisi et al. (2020) showed that the per-step variance $\mathbb{V}_{\pi_\theta}(R) \leq \frac{\mathbb{V}_{\pi_\theta}(r(S_t,A_t))}{(1-\gamma)^2}$, which implies that the minimiza-

tion of the per-step variance $\mathbb{V}_{\pi_\theta}(r(S_t, A_t))$ also minimizes trajectory-variance $\mathbb{V}_{\pi_\theta}(R)$. Therefore, they train a policy $\pi_\theta$ by maximizing $J_\kappa(\pi) = \mathbb{E}_{\pi_\theta}[r(S_t, A_t)] - \kappa\mathbb{V}_{\pi_\theta}(r(S_t, A_t))$, where $\kappa > 0$ is a parameter of the penalty function. Bisi et al. (2020) reveals that $J_\kappa(\pi) = \mathbb{E}_{\pi_\theta}[R - \kappa(R - \mathbb{E}_{\pi_\theta}[R])^2]$, which implies that optimizing $J_\kappa(\theta)$ is equivalent to optimize the canonical risk-neutral objective of a new MDP with the same as the original MDP except that the new reward function $r'(s, a) = r(s, a) - \lambda(r(s, a) - (1 - \gamma)J(\pi))^2$. This reward function depends on the policy $\pi$, making the reward function nonstationary and conventional RL method unusable. This problem is called policy-dependent-reward issue. To solve this policy-dependent-reward issue, the methods of Bisi et al. (2020) and Zhang et al. (2021b) are based on the trust region policy optimization (Schulman et al., 2015) and coordinate descent with Legendre-Fenchel duality (Xie et al., 2018), respectively.

For constants $\alpha, \beta > 0$, we define the quadratic utility function for per-step reward setting as

$$u^{\text{per-step}}(r(S_t, A_t); \alpha, \beta) = \alpha r(S_t, A_t) - \beta r^2(S_t, A_t).$$

For an infinite horizon case with $\gamma < 1$, we consider maximizing the following expected cumulative quadratic utility function:

$$\mathbb{E}_{\pi_\theta}\left[\sum_{t=1}^{\infty}\gamma^{t-1}u^{\text{per-step}}(r(S_t, A_t); \alpha, \beta)\right] = \alpha\mathbb{E}_{\pi_\theta}\left[\sum_{t=1}^{\infty}\gamma^{t-1}r(S_t, A_t)\right] - \beta\mathbb{E}_{\pi_\theta}\left[\sum_{t=1}^{\infty}\gamma^{t-1}r^2(S_t, A_t)\right]$$

For an finite horizon case with $\gamma = 1$, we consider maximizing the following expected cumulative quadratic utility function:

$$\mathbb{E}_{\pi_\theta}\left[\sum_{t=1}^{T}u^{\text{per-step}}(r(S_t, A_t); \alpha, \beta)\right] = \alpha\mathbb{E}_{\pi_\theta}\left[\sum_{t=1}^{T}r(S_t, A_t)\right] - \beta\mathbb{E}_{\pi_\theta}\left[\sum_{t=1}^{T}r^2(S_t, A_t)\right]$$

When applying REINFORCE-based algorithm to train an agent by maximizing the objective function, the sample approximation of the gradient, for instance, is given as

$$\widehat{\nabla}\mathbb{E}_{\pi_\theta}\left[\sum_{t=1}^{T}u^{\text{per-step}}(r(S_t^k, A_t^k); \alpha, \beta)\right]$$

$$= \alpha\sum_{t=1}^{T}r(S_t^k, A_t^k)\nabla_\theta\log\pi_\theta(S_t^k, A_t^k) - \beta\sum_{t=1}^{T}\left(r(S_t^k, A_t^k)\right)^2\nabla_\theta\log\pi_\theta(S_t^k, A_t^k)$$

We investigate the empirical performance using the same setting as the portfolio management experiment of Section 6.1. Let $\zeta$ be $\alpha/\beta$ in the trajectory EQUMRL, and $\rho$ be $\alpha/\beta$ in the per-step EQUMRL. Under both the trajectory and per-step EQUMRLs, we train policies with the REINFORCE-based algorithm. The other settings are identical to that in Section 6.1. As Figure 4 shows, under both the trajectory and per-step EQUMRLs, the REINFORCE-based algorithms train MV-efficient policies well for the parameters $\zeta$ and $\rho$.

## C AMERICAN-STYLE OPTION WITH A SYNTHETIC DATASET

An American-style option refers to a contract that we can execute an option right at any time before the maturity time $\tau$; that is, a buyer who bought a call option has a right to buy the asset with the strike price $K_{\text{call}}$ at any time; a buyer who bought a put option has a right to sell the with the strike price $K_{\text{put}}$ at any time.

In the setting of Tamar et al. (2014) and Xie et al. (2018), the buyer simultaneously buy call and put options, which have the strike price $K_{\text{call}} = 1.5$ and $K_{\text{put}} = 1.$, respectively. The maturity time is set as $\tau = 20$. If the buyer executes the option at time $t$, the buyer obtains a reward $r_t = \max(0, K_{\text{put}} - x_t) + \max(0, x_t - W_{\text{call}})$, where $x_t$ is an asset price. We set $x_0 = 1$ and define the stochastic process as follows: $x_t = x_{t-1}f_u$ with probability $0.45$ and $x_t = x_{t-1}f_d$ with probability $0.55$, where $f_u$ and $f_d$. These parameters follows Xie et al. (2018).

As well as Section 6.1, we compare the EQUMRL with policy gradient (EQUM) with the REIN-FORCE, Tamar, and Xie. The other settings are also identical to Section 6.1. We show performances under several hyperparameter in Figure 5 and CRs and Vars of some of their results in the upper table

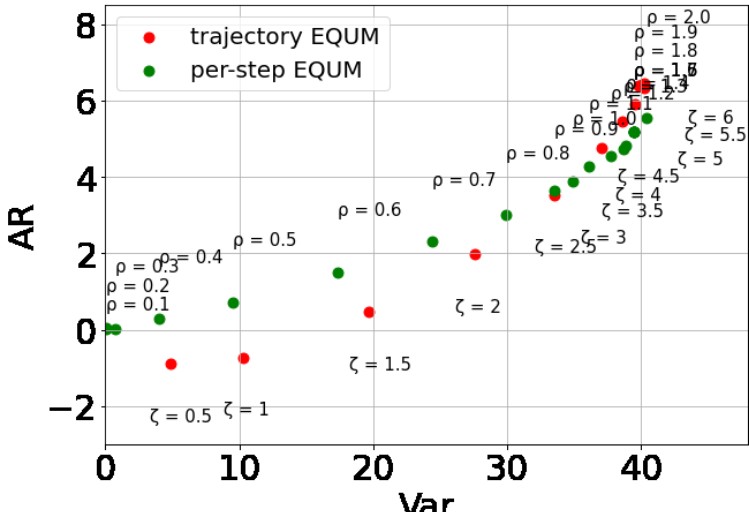

Figure 4: MV efficiency of the portfolio management experiment with the trajectory and per-step EQUMRLs. Higher CRs and lower Vars methods are MV Pareto efficient.

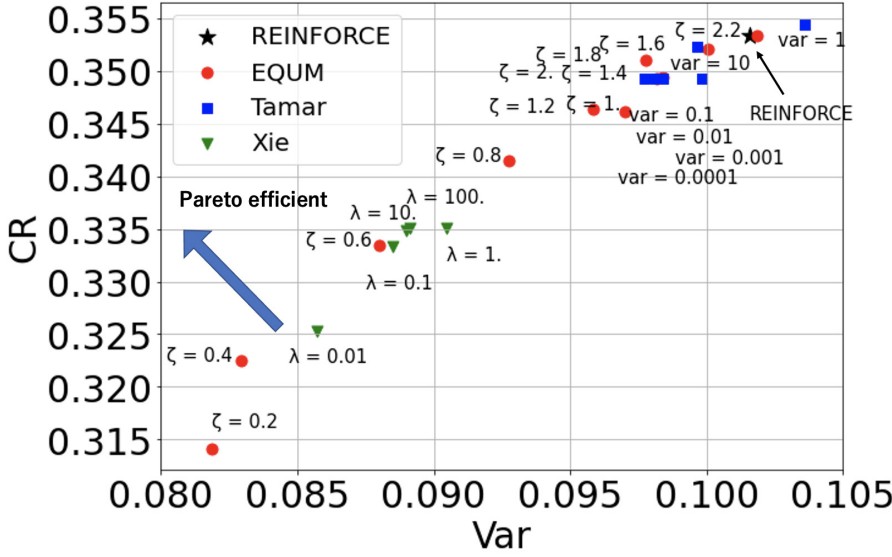

Figure 5: MV efficiency of the American-style option experiment. Higher CRs and lower Vars methods are MV Pareto efficient.

of Table 3. We also denote MSE between $\zeta$ and achieved CR of in the lower table of Table 3. From the table, we can find that while EQUMRL minimize the MSE for lower $\zeta$, the MSE of REINFORCE is smaller for higher $\zeta$. We consider this is because owing to the difficulty of MV control under this setting, naively maximizing the CR minimizes the MSE more than considering the MV trade-off for higher $\zeta$.

# D DETAILS OF EXPERIMENTS OF PORTFOLIO MANAGEMENT WITH A REAL-WORLD DATASET

We use the following portfolio models. An equally-weighted portfolio (EW) weights the financial assets equally (DeMiguel et al., 2009). A mean-variance portfolio (MV) computes the optimal variance under a mean constraint (Markowitz, 1952). To compute the mean vector and covariance matrix, we use the latest 10 years (120 months) data. A Kelly growth optimal portfolio with ensemble learning (EGO) is proposed by Shen et al. (2019). We set the number of resamples as

Table 3: The results of American-style option. The upper table shows the CRs and Vars over $100,000$ trials. The lower table shows The lower table shows the MSEs for $\zeta$ over $100,000$ trials.

| | REINFORCE | EQUM | | | Tamar | | Xie | |
|---|---|---|---|---|---|---|---|---|
| | ($\zeta = \infty$) | $\zeta = 1.6$ | $\zeta = 0.8$ | $\zeta = 0.4$ | $\mathbb{V} = 1$ | $\mathbb{V} = 0.1$ | $\lambda = 100$ | $\lambda = 0.1$ |
| CR | 0.353 | 0.352 | 0.341 | 0.322 | 0.352 | 0.349 | 0.335 | 0.333 |
| Var | 0.099 | 0.098 | 0.092 | 0.083 | 0.102 | 0.096 | 0.089 | 0.088 |

| | REINFORCE | EQUM | | | | |
|---|---|---|---|---|---|---|
| Target Value | ($\zeta = \infty$) | $\zeta = 2.2$ | $\zeta = 1.8$ | $\zeta = 1.4$ | $\zeta = 1.0$ | $\zeta = 0.6$ |
| MSE from $\zeta = 2.2$ | **3.512** | **3.512** | 3.517 | 3.523 | 3.532 | 3.572 |
| $\zeta = 1.8$ | **2.194** | 2.195 | 2.197 | 2.203 | 2.209 | 2.239 |
| $\zeta = 1.4$ | **1.197** | **1.197** | 1.198 | 1.202 | 1.206 | 1.226 |
| $\zeta = 1.0$ | 0.520 | 0.520 | **0.519** | 0.522 | 0.523 | 0.532 |
| $\zeta = 0.6$ | 0.162 | 0.163 | 0.160 | 0.161 | 0.160 | **0.159** |

$m_1 = 50$, the size of each resample $m_2 = 5\tau$, the number of periods of return data $\tau = 60$, the number of resampled subsets $m_3 = 50$, and the size of each subset $m_4 = n^{0.7}$, where $m$ is number of assets; that is, $m = 25$ in FF25, $m = 48$ in FF48 and $m = 100$ in FF100. Portfolio blending via Thompson sampling (BLD) is proposed by Shen & Wang (2016). We use the latest 10 years (120 months) data to compute for the sample covariance matrix and blending parameters. Denote a method proposed by Tamar et al. (2012) by Tamar, and choose the parameter var from $\{15, 30, 60\}$. Denote a method proposed by Xie et al. (2018) by Xie, and choose the parameter $\lambda$ from $\{10, 100, 1000\}$. Denote the REINFORCE-based trajectory EQUMRL by EQUMRL, and choose the parameter $\zeta$ from $\{0.5, 0.75, 1.5\}$. To train the policies of MVRL methods, we assume the stationarity on time-series. Given historical datasets $(y_{a,t})_{t \in \{1,...,T\}, a \in \{1,...,m\}}$, for each trajectory, we simulate portfolio management and generate $\{(S_t^k, A_t^k, r(S_t^k, A_t^k))\}_{t=1}^n$. Recall that state is given as $S_t = ((y_{a,t-1}, ..., y_{a,t-12}, w_{a,t-1})_{a \in \mathcal{A}}, \sum_{s=0}^{t-1} y_s)$, and the portfolio weight as given as $w_{a,t} = \pi(a, s_t)$ (see Section 6.2). Then, using the trajectory observations, we train the policies. We show the pseudo-code of the modified REINFORCE-based trajectory EQMRL in Algorithm 2.

---

**Algorithm 2** REINFORCE-based EQUMRL in Section 6.2

---

Historical dataset $(y_{a,t})_{t \in \{1,...,T\}, a \in \{1,...,m\}}$
Initialize the policy parameter $\theta_0$;
**for** $k = 1, 2, \ldots$ **do**
    Generate $\{(S_t^k, A_t^k, r(S_t^k, A_t^k))\}_{t=1}^n$ on $\pi_\theta$ by simulating the portfolio management using the historical dataset $(y_{a,t})_{t \in \{1,...,T\}, a \in \{1,...,m\}}$;
    Update policy parameters $\theta_{k+1} \leftarrow \theta_k + \eta_i \widehat{\nabla} \mathbb{E}_{\pi_{\theta_k}} \left[ u^{\text{trajectory}}(G; \alpha, \beta) \right]$.
**end for**

---

Table 4 shows the performance of each portfolio model without the penalty of the turnover $\Lambda = 0$. We report both results with and without the penalty of the turnover in the following. The average of real objective (minimizing variance with a penalty on return targeting) for Tamar, MVP and EQUMRL from July 2000 to June 2020 is shown in Table 5. We also divide the performance period into two for robustness checks. Table 6 and 7 shows the first-half results from July 2000 to June 2010 and the second-half results from July 2010 to June 2020. In almost all cases, the EQUMRL achieves the highest R/R.

We plot results of Tamar, Xie, and EQUMRL with various parameters in Figure 6. We choose the parameter of Tamar ($\mathbb{V}$) from $\{10, 15, 20, 25, 30, 35, 40, 45, 50, 55, 60, 65\}$, Xie ($\lambda$) from $\{10, 50, 100, 200, 300, 400, 500, 600, 700, 800, 900, 1000\}$, and EQUMRL ($\zeta$) from $\{0.125, 0.25, 0.375, 0.5, 0.625, 0.75, 0.875, 1, 1.125, 1.25, 1.375, 1.5\}$. We only annotate the points of $\mathbb{V} = 10, 25, 40, 55$, $\lambda = 10, 200, 500, 800$, and $\zeta = 0.125, 0.5, 0.875, 1.25$. Unlike Section 6.1, it is not easy to control the mean and variance owing to the difficulty of predicting the real-world financial markets. However, the EQUMRL tends to return more efficient results than the other methods.

Table 4: The performance of each portfolio model without the penalty of the turnover $\Lambda = 0$ during the out-of-sample period (from July 2000 to June 2020) for FF25 dataset (upper table) , FF48 (middle table), and FF100 (lower table). For each dataset, the best performance is highlighted in **bold**.

| Method | EW | MV | EGO | BLD | Tamar | | | Xie | | | EQUM | | |
|---|---|---|---|---|---|---|---|---|---|---|---|---|---|
| | | | | | $\mathbb{V}=15$ | $\mathbb{V}=30$ | $\mathbb{V}=60$ | $\lambda=10$ | $\lambda=100$ | $\lambda=1000$ | $\zeta=0.5$ | $\zeta=0.75$ | $\zeta=1.5$ |
| FF25 | | | | | | | | | | | | | |
| CR↑ | 0.80 | 0.11 | 0.87 | 0.56 | **1.16** | 1.00 | 0.80 | 0.96 | 1.01 | 0.90 | 1.11 | 1.12 | 1.02 |
| Var↓ | 28.62 | 53.58 | 30.63 | 12.01 | 18.41 | 15.10 | 25.22 | 24.94 | 14.98 | 18.37 | **11.77** | 21.29 | 16.25 |
| R/R↑ | 0.52 | 0.05 | 0.55 | 0.56 | 0.94 | 0.89 | 0.55 | 0.67 | 0.90 | 0.73 | **1.13** | 0.84 | 0.87 |
| MaxDD↓ | 0.54 | 0.75 | 0.57 | 0.37 | 0.35 | 0.35 | 0.55 | 0.46 | 0.34 | 0.49 | **0.27** | 0.33 | 0.30 |
| FF48 | | | | | | | | | | | | | |
| CR↑ | 0.81 | 0.15 | 1.04 | 0.52 | 0.90 | 0.63 | 1.06 | 0.52 | 0.34 | 0.35 | 0.92 | **1.49** | 1.01 |
| Var↓ | 22.91 | 76.89 | 31.87 | 9.65 | 40.13 | 15.81 | 18.07 | 11.79 | 6.72 | **6.70** | 15.11 | 30.96 | 27.28 |
| R/R↑ | 0.59 | 0.06 | 0.64 | 0.58 | 0.49 | 0.55 | 0.86 | 0.52 | 0.46 | 0.47 | 0.82 | **0.93** | 0.67 |
| MaxDD↓ | 0.30 | 0.48 | 0.31 | 0.18 | 0.32 | 0.19 | 0.21 | 0.20 | 0.17 | **0.17** | 0.20 | 0.27 | 0.28 |
| FF100 | | | | | | | | | | | | | |
| CR↑ | 0.81 | 0.14 | 0.86 | 0.53 | 1.07 | **1.86** | 1.36 | 1.33 | 1.05 | 1.07 | 1.35 | 1.30 | 1.38 |
| Var↓ | 29.36 | 57.90 | 32.40 | **11.79** | 31.61 | 85.74 | 20.05 | 43.65 | 20.94 | 87.24 | 20.62 | 16.56 | 27.50 |
| R/R↑ | 0.52 | 0.06 | 0.53 | 0.54 | 0.66 | 0.69 | 1.05 | 0.70 | 0.79 | 0.40 | 1.03 | **1.10** | 0.91 |
| MaxDD↓ | 0.33 | 0.46 | 0.34 | 0.22 | 0.29 | 0.57 | 0.22 | 0.36 | 0.23 | 0.51 | 0.23 | **0.20** | 0.28 |

Table 5: The average of real objective (minimizing variance with a penalty on return targeting) for Tamar, MVP and EQUMRL for FF25 dataset (upper panel) , FF48 (middle panel) and FF100 (lower panel) from July 2020 to June 2020.

| | Tamar | | | MVP | | | EQUM | | |
|---|---|---|---|---|---|---|---|---|---|
| | $\mathbb{V}=15$ | $\mathbb{V}=30$ | $\mathbb{V}=60$ | $\lambda=10$ | $\lambda=100$ | $\lambda=1000$ | $\zeta=0.5$ | $\zeta=0.75$ | $\zeta=1.5$ |
| With Turnover Penalty $\Lambda = 0.001$ | | | | | | | | | |
| FF25 | -5,466 | -7,504 | -7,646 | -6,227 | -6,398 | -7,769 | -7,853 | -8,758 | -9,027 |
| FF48 | -5,395 | -7,399 | -7,614 | -6,647 | -7,763 | -7,677 | -9,000 | -8,436 | -7,743 |
| FF100 | -6,711 | -7,491 | -6,233 | -6,362 | -5,354 | -6,983 | -8,681 | -7,366 | -8,362 |
| Without Turnover Penalty $\Lambda = 0$ | | | | | | | | | |
| FF25 | -5,312 | -7,486 | -7,329 | -6,130 | -6,390 | -7,340 | -7,664 | -7,660 | -8,911 |
| FF48 | -5,464 | -7,053 | -7,505 | -6,489 | -9,763 | -7,506 | -8,729 | -8,456 | -7,421 |
| FF100 | -6,749 | -7,287 | -5,915 | -6,047 | -5,446 | -7,008 | -9,484 | -7,306 | -8,280 |

Table 6: The performance of each portfolio without turnover penalty ($Lambda = 0$) during first half out-of-sample period (from July 2000 to June 2010) and second half out-of-sample period (from July 2010 to June 2020) for FF25 dataset (upper panel) , FF48 (middle panel) and FF100 (lower panel). Among the comparisons of the various portfolios, the best performance within each dataset is highlighted in **bold**.

| FF25 | EW | MV | EGO | BLD | Tamar | | | Xie | | | EQUM | | |
|---|---|---|---|---|---|---|---|---|---|---|---|---|---|
| | | | | | $\mathbb{V}=15$ | $\mathbb{V}=30$ | $\mathbb{V}=60$ | $\lambda=10$ | $\lambda=100$ | $\lambda=1000$ | $\zeta=0.5$ | $\zeta=0.75$ | $\zeta=1.5$ |
| First-Half Period (from July 2000 to June 2010) | | | | | | | | | | | | | |
| CR↑ | 0.58 | -0.42 | 0.64 | 0.41 | 1.35 | 1.31 | 1.18 | 0.96 | 1.32 | 1.15 | **1.54** | 1.34 | 1.22 |
| Var↓ | 31.21 | 69.36 | 33.29 | 12.50 | 14.47 | 11.34 | 11.78 | 13.67 | 10.99 | 11.50 | **9.96** | 22.79 | 17.10 |
| R/R↑ | 0.36 | -0.17 | 0.38 | 0.41 | 1.23 | 1.35 | 1.19 | 0.72 | 1.38 | 1.18 | **1.69** | 0.97 | 1.02 |
| MaxDD↓ | 0.54 | 0.75 | 0.58 | 0.37 | 0.21 | 0.16 | 0.19 | 0.32 | 0.15 | **0.19** | 0.10 | 0.33 | 0.21 |
| Second-Half Period (from July 2000 to June 2010) | | | | | | | | | | | | | |
| CR↑ | 1.02 | 0.63 | **1.03** | 0.70 | 0.98 | 0.68 | 0.43 | 0.96 | 0.70 | 0.65 | 0.69 | 0.90 | 0.81 |
| Var↓ | 25.92 | 37.24 | 27.92 | **11.47** | 22.30 | 18.66 | 38.39 | 28.50 | 18.77 | 25.10 | 13.21 | 19.69 | 15.32 |
| R/R↑ | 0.69 | 0.36 | 0.67 | 0.72 | 0.72 | 0.55 | 0.24 | 0.62 | 0.56 | 0.45 | 0.66 | 0.70 | **0.72** |
| MaxDD↓ | 0.31 | 0.50 | 0.31 | **0.22** | 0.35 | 0.35 | 0.55 | 0.46 | 0.34 | 0.49 | 0.27 | 0.32 | 0.30 |

| FF48 | EW | MV | EGO | BLD | Tamar | | | Xie | | | EQUM | | |
|---|---|---|---|---|---|---|---|---|---|---|---|---|---|
| | | | | | $\mathbb{V}=15$ | $\mathbb{V}=30$ | $\mathbb{V}=60$ | $\lambda=10$ | $\lambda=100$ | $\lambda=1000$ | $\zeta=0.5$ | $\zeta=0.75$ | $\zeta=1.5$ |
| First-Half Period (from July 2000 to June 2010) | | | | | | | | | | | | | |
| CR↑ | 0.60 | 0.41 | 0.75 | 0.34 | 1.57 | 0.68 | 1.35 | 1.03 | -0.25 | -0.26 | 1.23 | **1.72** | 1.09 |
| Var↓ | 25.85 | 57.50 | 39.33 | 10.97 | 41.06 | 14.46 | 13.85 | 7.07 | 0.02 | **0.02** | 11.15 | 27.16 | 18.30 |
| R/R↑ | 0.41 | 0.19 | 0.41 | 0.36 | 0.85 | 0.62 | 1.26 | **1.34** | -6.36 | -6.30 | 1.28 | 1.15 | 0.88 |
| MaxDD↓ | 0.27 | 0.33 | 0.31 | 0.18 | 0.30 | 0.17 | 0.17 | 0.17 | 0.00 | **0.00** | 0.16 | 0.27 | 0.18 |
| Second-Half Period (from July 2000 to June 2010) | | | | | | | | | | | | | |
| CR↑ | 1.02 | -0.12 | **1.33** | 0.69 | 0.23 | 0.59 | 0.77 | 0.01 | 0.94 | 0.97 | 0.61 | 1.27 | 0.92 |
| Var↓ | 19.88 | 96.14 | 24.24 | **8.27** | 38.29 | 17.16 | 22.13 | 16.00 | 12.72 | 12.63 | 18.87 | 34.66 | 36.24 |
| R/R↑ | 0.79 | -0.04 | 0.93 | 0.84 | 0.13 | 0.49 | 0.57 | 0.01 | 0.91 | **0.94** | 0.48 | 0.75 | 0.53 |
| MaxDD↓ | 0.27 | 0.48 | 0.27 | 0.28 | 0.28 | 0.19 | 0.21 | 0.17 | 0.17 | **0.17** | 0.20 | 0.26 | 0.28 |

| FF100 | EW | MV | EGO | BLD | Tamar | | | MVP | | | EQUM | | |
|---|---|---|---|---|---|---|---|---|---|---|---|---|---|
| | | | | | $\mathbb{V}=15$ | $\mathbb{V}=30$ | $\mathbb{V}=60$ | $\lambda=10$ | $\lambda=100$ | $\lambda=1000$ | $\zeta=0.5$ | $\zeta=0.75$ | $\zeta=1.5$ |
| First-Half Period (from July 2000 to June 2010) | | | | | | | | | | | | | |
| CR↑ | 0.61 | -0.37 | 0.73 | 0.41 | 1.08 | 1.88 | 1.67 | 1.42 | 1.29 | 0.73 | 1.43 | 1.50 | **1.93** |
| Var↓ | 32.02 | 79.35 | 35.11 | **12.43** | 19.67 | 109.44 | 14.33 | 40.80 | 13.16 | 92.80 | 17.16 | 12.80 | 21.45 |
| R/R↑ | 0.37 | -0.15 | 0.43 | 0.40 | 0.84 | 0.62 | **1.52** | 0.77 | 1.23 | 0.26 | 1.20 | 1.45 | 1.44 |
| MaxDD↓ | 0.28 | 0.46 | 0.30 | 0.17 | 0.22 | 0.52 | 0.18 | 0.30 | **0.15** | 0.51 | 0.20 | 0.17 | 0.22 |
| Second-Half Period (from July 2000 to June 2010) | | | | | | | | | | | | | |
| CR↑ | 1.01 | 0.66 | 1.00 | 0.66 | 1.06 | **1.83** | 1.06 | 1.24 | 0.80 | 1.41 | 1.27 | 1.09 | 0.83 |
| Var↓ | 26.61 | 35.92 | 29.64 | **11.12** | 43.56 | 62.05 | 25.58 | 46.48 | 28.61 | 81.45 | 24.07 | 20.23 | 32.94 |
| R/R↑ | 0.68 | 0.38 | 0.63 | 0.68 | 0.56 | 0.81 | 0.73 | 0.63 | 0.52 | 0.54 | **0.90** | 0.84 | 0.50 |
| MaxDD↓ | 0.32 | 0.33 | 0.32 | 0.22 | 0.27 | 0.34 | 0.22 | 0.33 | 0.23 | 0.45 | 0.22 | **0.19** | 0.27 |

Table 7: The performance of each portfolio with turnover penalty ($\Lambda = 0.001$) during first half out-of-sample period (from July 2000 to June 2010) and second half out-of-sample period (from July 2010 to June 2020) for FF25 dataset (upper panel) , FF48 (middle panel) and FF100 (lower panel). Among the comparisons of the various portfolios, the best performance within each dataset is highlighted in **bold**.

| FF25 | EW | MV | EGO | BLD | Tamar | | | Xie | | | EQUM | | |
|---|---|---|---|---|---|---|---|---|---|---|---|---|---|
| | | | | | $\mathbb{V}=2.5$ | $\mathbb{V}=15$ | $\mathbb{V}=30$ | $\lambda=60$ | $\lambda=100$ | $\lambda=1000$ | $\zeta=0.5$ | $\zeta=0.75$ | $\zeta=1.5$ |
| First-Half Period(from July 2000 to June 2010) | | | | | | | | | | | | | |
| CR↑ | 0.58 | -0.43 | 0.64 | 0.56 | 1.32 | 1.31 | 1.19 | 0.93 | 1.17 | 1.17 | **1.58** | 1.55 | 1.54 |
| Var↓ | 31.21 | 69.33 | 33.29 | 25.08 | 14.46 | 11.18 | 11.81 | 11.95 | 11.93 | 11.93 | 16.47 | 11.18 | **9.96** |
| R/R↑ | 0.36 | -0.18 | 0.38 | 0.39 | 1.21 | 1.36 | 1.20 | 0.93 | 1.17 | 1.17 | 1.35 | 1.61 | **1.69** |
| MaxDD↓ | 0.54 | 0.75 | 0.58 | 0.52 | 0.21 | 0.15 | 0.19 | 0.20 | 0.17 | 0.17 | 0.26 | 0.13 | **0.10** |
| Second-Half Period(from July 2000 to June 2010) | | | | | | | | | | | | | |
| CR↑ | 1.02 | 0.60 | 0.99 | 1.12 | 0.97 | 0.70 | 0.41 | 0.87 | 1.13 | 1.13 | **1.48** | 0.95 | 0.69 |
| Var↓ | 25.92 | 37.28 | 27.95 | 19.09 | 22.54 | 18.72 | 38.46 | 39.97 | 39.61 | 39.61 | 32.08 | 18.67 | **13.21** |
| R/R↑ | 0.69 | 0.34 | 0.65 | 0.89 | 0.71 | 0.56 | 0.23 | 0.48 | 0.62 | 0.62 | **0.90** | 0.76 | 0.66 |
| MaxDD↓ | 0.31 | 0.52 | 0.32 | **0.25** | 0.35 | 0.35 | 0.56 | 0.54 | 0.51 | 0.51 | 0.36 | 0.31 | 0.27 |
| FF48 | EW | MV | EGO | BLD | Tamar | | | Xie | | | EQUM | | |
| | | | | | $\mathbb{V}=2.5$ | $\mathbb{V}=15$ | $\mathbb{V}=30$ | $\lambda=60$ | $\lambda=100$ | $\lambda=1000$ | $\zeta=0.5$ | $\zeta=0.75$ | $\zeta=1.5$ |
| First-Half Period(from July 2000 to June 2010) | | | | | | | | | | | | | |
| CR↑ | 0.60 | 0.36 | 0.70 | 0.49 | 1.36 | 0.41 | 1.11 | 0.58 | 1.21 | 0.82 | 1.35 | 1.18 | **1.36** |
| Var↓ | 25.85 | 57.76 | 39.39 | 18.69 | 41.27 | 14.49 | 13.70 | 18.51 | 17.61 | 19.28 | 29.97 | 20.19 | **6.04** |
| R/R↑ | 0.41 | 0.16 | 0.38 | 0.39 | 0.73 | 0.37 | 1.04 | 0.47 | 1.00 | 0.64 | 0.85 | 0.91 | **1.92** |
| MaxDD↓ | 0.27 | 0.33 | 0.31 | 0.25 | 0.30 | 0.18 | 0.17 | 0.18 | 0.20 | 0.19 | 0.25 | 0.17 | **0.10** |
| Second-Half Period(from July 2000 to June 2010) | | | | | | | | | | | | | |
| CR↑ | 1.02 | -0.14 | 1.25 | 1.02 | -0.00 | 0.36 | 0.52 | 0.43 | 0.94 | 1.20 | **1.86** | 0.92 | 0.45 |
| Var↓ | 19.88 | 96.16 | 24.28 | **13.14** | 39.17 | 17.52 | 22.21 | 35.72 | 29.90 | 33.27 | 33.85 | 17.15 | 14.22 |
| R/R↑ | 0.79 | -0.05 | 0.88 | 0.98 | -0.00 | 0.29 | 0.38 | 0.25 | 0.59 | 0.72 | **1.11** | 0.77 | 0.41 |
| MaxDD↓ | 0.27 | 0.48 | 0.28 | 0.22 | 0.28 | 0.20 | 0.21 | 0.27 | 0.25 | 0.26 | 0.27 | 0.20 | **0.16** |
| FF100 | EW | MV | EGO | BLD | Tamar | | | MVP | | | EQUM | | |
| | | | | | $\mathbb{V}=2.5$ | $\mathbb{V}=15$ | $\mathbb{V}=30$ | $\lambda=60$ | $\lambda=100$ | $\lambda=1000$ | $\zeta=0.5$ | $\zeta=0.75$ | $\zeta=1.5$ |
| First-Half Period(from July 2000 to June 2010) | | | | | | | | | | | | | |
| CR↑ | 0.61 | -0.41 | 0.67 | 0.59 | 1.07 | 0.98 | 1.23 | 0.24 | 1.26 | **1.39** | 1.19 | 1.17 | 1.22 |
| Var↓ | 32.02 | 79.44 | 35.06 | 24.25 | 14.20 | 13.92 | 11.39 | 93.47 | 16.00 | 28.49 | 13.42 | **9.16** | 17.63 |
| R/R↑ | 0.37 | -0.16 | 0.39 | 0.41 | 0.99 | 0.91 | 1.26 | 0.09 | 1.09 | 0.90 | 1.12 | **1.34** | 1.01 |
| MaxDD↓ | 0.28 | 0.46 | 0.30 | 0.25 | 0.17 | 0.18 | 0.18 | 0.51 | 0.21 | 0.21 | 0.17 | **0.15** | 0.22 |
| Second-Half Period(from July 2000 to June 2010) | | | | | | | | | | | | | |
| CR↑ | 1.01 | 0.63 | 0.95 | 1.12 | 1.18 | 0.60 | 0.11 | 0.90 | 0.70 | 1.09 | 0.70 | 0.72 | **1.64** |
| Var↓ | 26.61 | 35.96 | 29.61 | 19.27 | 36.80 | 26.69 | 20.99 | 82.15 | 41.40 | 53.59 | **17.46** | 19.26 | 48.45 |
| R/R↑ | 0.68 | 0.37 | 0.60 | **0.88** | 0.67 | 0.40 | 0.08 | 0.34 | 0.38 | 0.52 | 0.58 | 0.57 | 0.82 |
| MaxDD↓ | 0.32 | 0.33 | 0.32 | 0.26 | 0.25 | 0.23 | 0.22 | 0.45 | 0.32 | 0.31 | **0.19** | 0.26 | 0.31 |

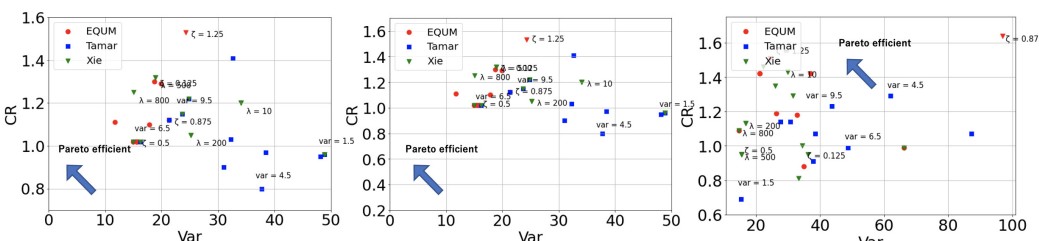

Figure 6: MV efficiency of the portfolio management experiment. Higher CRs and lower Vars methods are MV Pareto efficient. Left graph: FF25. Center graph: FF48. Right graph: FF100.

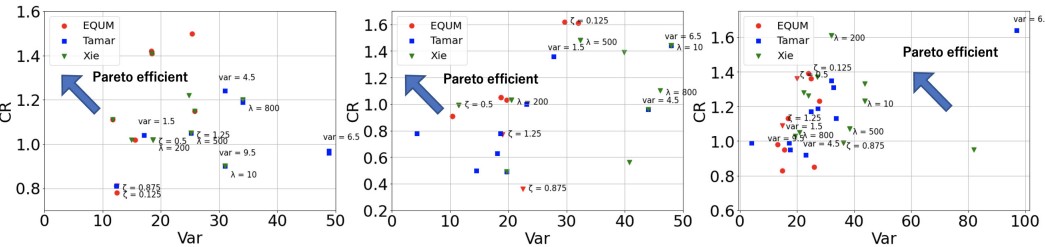

Figure 7: MV efficiency of the portfolio management experiment. Higher CRs and lower Vars methods are MV Pareto efficient. Left graph: FF25. Center graph: FF48. Right graph: FF100.

