# OpenReview forum: "Mean-Variance Efficient Reinforcement Learning by Expected Quadratic Utility Maximization"
_ICLR.cc/2022/Conference — ICLR 2022 Submitted_

### Official Review · Reviewer_TnBm · 2021-11-01

**Correctness:** 3
**Technical Novelty And Significance:** 2
**Empirical Novelty And Significance:** 2
**Recommendation:** 5
**Confidence:** 4

**Main Review:**

Strengths: Authors use well known results from mathematical finance to improve state-of-the-art for portfolio optimization using reinforcement learning. The paper is also well written and the flow is clear.

Weakness: I have concerns on results presented for both the examples illustrated in the paper. In addition, authors need to improve the writing and update literature review to provide better context of the contribution of this work. Please see below for detailed comments:

*Contribution/Results:*
1. It is not clear why the proposed algorithm works better than the Tamar et. al. and Xie et. al. Author’s state: “We conjecture that this is because while the EQUMRL is an end-to-end optimization for obtaining an efficient agent, the other methods consist of several steps for solving the constrained optimization, where those multiple steps can be sources of the suboptimal result.”. This provides no concrete explanation or evidence on the difference.
2. Results in Figure 2 are somewhat concerning. It is not clear why target variance is much higher than the observed variance (e.g., blue square with “var=50” has an observed variance of 33). Since returns increase with increasing variance, I would expect the constraint to be binding.
3. For the real world dataset, it is not clear how the authors have trained the model. Since there is only one historical trajectory of each stock in the portfolio, it is not obvious how the training works. It will be helpful to provide some details on model training in this context.

*Writing:*
1. It will be helpful if authors could clearly set up the portfolio optimization problem in Section 2. The current description is very generic. As an example, they do not describe what is the action space (e.g., portfolio weights or actual dollar investments), the state space (returns or stock prices) or the rewards for the problem considered in the paper. It only starts to become somewhat clear when the reader is in Section 6. There are a lot of different ways one can set up the portfolio optimization problem, so providing clarity on it is important.
2. Authors do not describe the sequence of steps required for implementing the algorithm. The practitioners will find the paper far more useful if at least it could provide the complete algorithm at one place.
3. The discussion on per-step variance perspective has no corresponding results. Authors should probably consider moving it as a discussion topic at the end or provide additional results to support the setup.

*Literature:*
There is a rich literature (over 20 years of work) on portfolio optimization in mathematical finance using stochastic control which also solves for dynamic decision problems and is closely related to the work done by the authors. Providing some context of how this work relates to that domain would be helpful.

*Typos:*
1. Section 6, first line “variacen” → “variance”
2. Page 1, 4th last line - should “Pareto efficient policy” be “pareto efficient frontier”?


**Summary Of The Paper:**

In this paper, authors propose a reinforcement learning approach to solve mean-variance portfolio optimization problem. The main idea of the paper relies on transforming mean variance optimization to maximizing expected quadratic utility. This allows the authors to avoid inefficient approximation of the gradient of the variance and instead use the same sample paths to approximate the gradient of the first and second moments of the reward. They implement two algorithms, first based on REINFORCE and another based on Actor-Critic method. Authors also illustrate the performance of the method on two problems, first based on synthetic dataset and another based on Fama & French dataset.

**Summary Of The Review:**

My recommendation is to reject because the results require more justification on the performance of the method relative to state-of-the-art. The work does not provide any novel perspective over what is already known to the financial mathematics or reinforcement learning community. Authors need to provide more clarity in writing, In particular, in formulating the optimization problem and how the method is implemented.

---

> ### Author Response · Authors · 2021-11-23
> **Response to Reviewer TnBm**
>
> We appreciate your constructive comments. In the revised manuscript, we fixed the typos and continue to reflect your comments on our manuscript.
>
> Firstly, we would like to clarify that our work is not on portfolio management or finance, although we consider them as important applications of our proposed method, as well as existing studies Tamar et al.
> > Literature: There is a rich literature (over 20 years of work) on portfolio optimization in mathematical finance using stochastic control, which also solves for dynamic decision problems and is closely related to the work. Providing some context of how this work relates to that domain would be helpful.
>
> In Appendix A, we explain the background in finance.  Although the reviewer suggests a more recent literature review, and we will add them as possible at the camera-ready, we consider that most of such financial studies are less relevant to the context of RL. For example, some studies assume a stochastic process for financial assets and obtain analytical solutions, which is a different formulation from RL. The goal of our work is to apply the classical result known in finance to improve the methods proposed in the RL literature, where this classic result is not well known.
>
> Our replies to your comments are listed below.
>
> **Q1**
> It is not clear why the proposed algorithm works better than the Tamar et al. and Xie et al. Author’s state: “We conjecture that this is because while the EQUMRL is an end-to-end optimization for obtaining an efficient agent, the other methods consist of several steps for solving the constrained optimization, where those multiple steps can be sources of the suboptimal result.“. This provides no concrete explanation or evidence on the difference.
>
> **A1**
> Xie et al. formulates the MVRL with a constrained optimization problem, but the method cannot consider the constraint because the constraint is not used in optimization. As a result, the objective is the same as ours, the expected quadratic utility function. However, their method includes complicated steps, which may not be necessary, as we pointed out.
>
> Tamar et al. applies computationally difficult steps in optimization, multi-time-scale stochastic optimization, which requires control of learning parameters. This difficulty of this optimization is well-known in RL, as pointed out by Xie et al.
>
> While we point out the computational difficulties of the method of Tamar et al., which is also mentioned in the other existing studies, we avoid explicitly mentioning the problem in Xie et al., as that would be a discussion of the correctness of that paper.
>
> **Q2**
> Results in Figure 2 are somewhat concerning. It is not clear why target variance is much higher than the observed variance (e.g., blue square with “var=50” has an observed variance of 33). Since returns increase with increasing variance, I would expect the constraint to be binding.
>
> **A2**
> The method is proposed by Tamar et al., not by us, and this is the reason why we proposed our method. It is quite difficult to solve constrained optimization with some computational issues. As explained above and by Xie et al., Tamar et al. still has a computational difficulty, and we conjecture that the method also cannot control the variance exactly. Thus, the problem that the reviewer is concerned about is not due to our implementation but to the methods of Tamar et al. While Tamar et al.’s theoretical and methodological contributions are significant, there is a room for improvement in its empirical performance.
>
> **Q3**
> For the real-world dataset, it is not clear how the authors have trained the model.
>
> **A3**
> We assume the stationary of the time series and train the policy using resampled observations, as we show it in Algorithm 2 of the updated manuscript. At the camera-ready, we add a more detailed description.
>
> **Q4**
> It would be helpful if authors could clearly set up the portfolio optimization problem in Section 2. The current description is very generic. As an example, they do not describe what is the action space (e.g., portfolio weights or actual dollar investments), the state space (returns or stock prices) or the rewards for the problem considered in the paper.
>
> **A4**
> In the revised manuscript, we add more descriptions on the MDP. Since this work considers RL, not finance, we avoided introducing the portfolio setting in Section 2.
>
> **Q5**
> Authors do not describe the sequence of steps required for implementing the algorithm.
>
> **A5**
> Thank you for your suggestion. We add the pseudo-code of REINFORCE-based trajectory EQUMRL (Algorithm 1) in Section 4.2.
>
> **Q6**
> The discussion on per-step variance perspective has no corresponding results. Authors should probably consider moving it as a discussion topic at the end or provide additional results to support the setup.
>
> **A6**
> In Appendix B of the updated manuscript, we have added the results of simple experiments.

---

### Official Review · Reviewer_MBfd · 2021-11-02

**Correctness:** 4
**Technical Novelty And Significance:** 3
**Empirical Novelty And Significance:** 3
**Recommendation:** 8
**Confidence:** 4

**Main Review:**

The paper is well written and easy to follow in general. The survey in the appendix is helpful for understanding concepts and motivations from a finance and economics perspective. The authors provide detailed discussion of the related literature and comparison of the present work with existing ones. They also present ample experimental results under various settings (related to investment) to justify the efficacy of the algorithms.

Questions:
- pp4, line 7, E[G] + \lambda (...) should be E[G] - \lambda (...)?
- pp5, line 4, \hat{G}^k should approximate E[G]?
- pp5, the displayed equation seems to imply that (\hat{G}^k)^2 is an unbiased estimateor of E[G^2]. Can the authors provide intuition of why this is the case?
- I am curious of how the actor-critic implementation would fare against the REINFORCE-based method in experiments.


---- Post rebuttal ----

I appreciate the authors' response. I have read through other reviewers' comments and authors' rebuttal. I would like to maintain my score.

**Summary Of The Paper:**

The paper studies the reinforcement learning problem with mean-variance tradeoff. It investigates a quadratic formulation that aims to optimize for the mean-variance efficiency, in a sense similar to the Pareto efficiency. It establishes connections between the quadratic formulation with the standard mean-variance formulation studied in the literature, and provides various interpretations of the mean-variance efficiency. The paper also proposes novel algorithmic implementations of the quadratic formulation based on REINFORCE and actor-critic methods. Three investment-related experiments are performed for the REINFORCE-based algorithm and demonstrate its good performance compared with existing algorithms.

**Summary Of The Review:**

The paper considers an important problem and its proposed algorithm is justified by extensive experiment results.

---

> ### Author Response · Authors · 2021-11-20
> **Response to Reviewer MBfd**
>
> Thank you for your comments and positive feedback.
>
> Our replies to your comments are listed below.
>
> ------------
> **Q1**
> - pp4, line 7, $E[G] + \lambda (...)$ should be $E[G] - \lambda (...)$?
> - pp5, line 4, $\hat{G}^k$ should approximate $E[G]$?
>
> **A1**
> Thank you for pointing them out. As you pointed out, they are our typos. We fixed them in the updated manuscript.
>
> ------------
> **Q2**
> pp5, the displayed equation seems to imply that $(\hat{G}^k)^2$ is an unbiased estimateor of $E[G^2]$. Can the authors provide intuition of why this is the case?
>
> **A2**
> Let $P$ be the probability measure, and $G = \sum^n_{t=1}\gamma^{t-1}r(S_t, A_T)$ be a random variable on the measure. Then, the second moment $E[G^2]$ is rewritten as
> $$ \int G^2 dP = \int \left(\sum^n_{t=1}\gamma^{t-1}r(S_t, A_T)\right)^2 dP.$$
> This directly implies $(\hat{G}^k)^2$ is an unbiased estimate of $E[G^2]$.
>
> ------------
>
> **Q3**
> I am curious of how the actor-critic implementation would fare against the REINFORCE-based method in experiments.
>
> **A3**
> Thanks for the suggestion. We will add the results in a simple example during rebuttal or in the camera-ready.

---

### Official Review · Reviewer_fgBH · 2021-11-02

**Correctness:** 3
**Technical Novelty And Significance:** 3
**Empirical Novelty And Significance:** Not applicable
**Recommendation:** 6
**Confidence:** 4

**Main Review:**

Strengths:
- The paper is nicely written with a clear motivation, i.e., to resolve the computational/sampling issues for traditional methods that aim to optimize the mean-variance objective.
- The proposed expected quadratic utility objective is easy to work with.
- The interpretation of the proposed objective is well-explained.

Weaknesses/Questions:
- Is it possible to provide some quantifiable gap between the optimal policy under the proposed objective and the optimal policy under the original mean-variance objective?
- In the experiments, it seems that only the mean performances are shown. What are the standard deviations for the performances of the methods?
- side points: (1) I am not sure how the traditional MVRL objectives can be rephrased as the constraint optimization problem with *equality* constraint. Could the authors provide more details on that? (2) The \nabla_{REINFORCE} notation seems very unconventional.



**Summary Of The Paper:**

The paper proposed an alternative objective (expected quadratic utility) to the mean-variance objective in an episodic RL setting. The new objective does not involve the squared expectation term, thus resolving the double sampling issue for obtaining the gradient of the mean-variance objective. In addition, the new objective has a couple of nice interpretations including the Pareto-efficiency one.

**Summary Of The Review:**

The paper was nicely written and provided a simple alternative (the expected quadratic utility function) to the mean-variance objective, which has nice interpretations and avoids the double sampling issues.

---

> ### Author Response · Authors · 2021-11-19
> **Response to Reviewer fgBH**
>
> Thank you for your constructive comments. We revised our manuscript based on your feedback.
>
> Our replies to your comments are listed below.
>
> ----------------
>
> **Q1**.
> Is it possible to provide some quantifiable gap between the optimal policy under the proposed objective and the optimal policy under the original mean-variance objective?
>
> **A1**
> Does the original mean-variance objective mean constrained optimization (MV-controlled problem in our paper)? The optimal policies of the constrained optimization problem are located on the Pareto efficient frontier in the sense of MV trade-off (MV efficient frontier). The optimal policy under the proposed EQUM is one of the points on the MV efficient frontier. Thus, there is no quantifiable gap between the optimal policy under the proposed objective and the optimal policy under the original mean-variance objective if a task and hyper-parameters are well-defined.
>
> While we do not aim to control the variance (or mean) at a certain level, we reduce the computational costs by avoiding the double sampling issue. Note that although the concepts are different, existing methods based on the constrained problem also do not control the variance (or mean) at a certain level because solving the constrained problem is computationally intractable. To convert our proposed method to the MV-controlled problem, for instance, we can derive several solutions with our method under various hyper-parameters and choose the most desirable solution among the solutions (see Remark 4).  Also, see Figure 3 in Appendix A for the relationships between them in finance.
>
> ----------------
>
> **Q2**
> In the experiments, it seems that only the mean performances are shown. What are the standard deviations for the performances of the methods?
>
> **A2**
> For each experiment, we show the variance with the mean performances, and the squared root of the variance corresponds to the standard deviation of the method.
>
> ----------------
>
> **Q3**
> side points:
>
> **Q3 (1)**
> I am not sure how the traditional MVRL objectives can be rephrased as the constraint optimization problem with equality constraint. Could the authors provide more details on that?
>
> **Q3 (2)**
> It is possible to consider the case where the constraints are not satisfied by equality. However, if the constraints and variances are chosen appropriately, it is more natural to satisfy them by equality. This is because when the optimal solution does not satisfy the equality, the solution is the same as the expected reward maximization problem without considering the variance (unconstrained problem). In fact, in finance and OR, equality constrained optimization is used typically. Therefore, in this manuscript, we only consider equality constraints as well.
>
>
> **Q3 (2)**
> The \nabla_{REINFORCE} notation seems very unconventional.
>
> **A3 (2)**
> Thank you for your comments! We remove it in the update manuscript.

---

### Official Review · Reviewer_wFpo · 2021-11-02

**Correctness:** 3
**Technical Novelty And Significance:** 2
**Empirical Novelty And Significance:** 3
**Recommendation:** 5
**Confidence:** 3

**Main Review:**

Strengths:

1. The studied mean-variance problem is very interesting and well-motivated.
2. This paper conducts extensive experimental results and demonstrate the empirical performance superiority of the proposed EQUMRL.
3. The authors provide code, which is helpful for readers to reproduce their results.

Weakness:

This paper lacks theoretical analysis. In particular, what regret/mean-covariance/Pareto efficiency guarantee can the proposed EQUMRL achieve? I expect to see formal theorems and rigorous proofs.


----After Rebuttal----

I read the authors' response. In general, I think that the weakness of this paper falls on theoretical analysis. Although the authors explained that the regret analysis in RL is an open problem, to my best knowledge, there are a number of papers that study regret bounds for (online) RL under tabular or linear MDP settings in recent years.

For example,

Azar, M. G., Osband, I. and Munos, R. (2017). Minimax regret bounds for reinforcement
learning. In Proceedings of the 34th International Conference on Machine Learning-Volume 70.
JMLR. org.

Jin, C., Allen-Zhu, Z., Bubeck, S. and Jordan, M. I. (2018). Is Q-learning provably efficient?
In Advances in Neural Information Processing Systems.

Jin, C., Yang, Z., Wang, Z. and Jordan, M. I. (2020). Provably efficient reinforcement learning with linear function approximation. In Conference on Learning Theory.

I more or less understand that it is hard to analyze a policy gradient based RL algorithm with sophisticated parametric models like neural networks. But the authors' response did not show me sufficient contributions of this paper.

In my opinion, this paper lacks solid theoretical analysis, and its experiment part looks standard (does not provide empirical evaluations under sufficiently comprehensive and challenging experimental benchmarks/setups). I did not find enough reasons to recommend accept.
So I plan to stick to my score 5.

**Summary Of The Paper:**

This paper learns MV-efficient policies that achieve Pareto efficiency in terms of MV trade-off. The authors propose an approach called expected quadratic utility maximization (EQUMRL), which trains an agent to maximize the expected quadratic utility function. The proposed EQUMRL does not suffer from the computational difficulties. Experimental results are provided to demonstrate the effectiveness of the proposed EQUMRL.

**Summary Of The Review:**

Overall, I think that the studied mean-covariance problem in this paper is very interesting. The idea behind the proposed approach looks non-trivial. Extensive experimental results are provided to show the superior empirical performance. However, this paper lacks theoretical analysis, and thus I could not judge whether the proposed approach achieves good theoretical results as well.

---

> ### Author Response · Authors · 2021-11-19
> **Response to Reviewer wFpo**
>
> Thank you for your insightful suggestion. We answer your comment as follows.
>
> **Q**
> This paper lacks theoretical analysis. In particular, what regret/mean-covariance/Pareto efficiency guarantee can the proposed EQUMRL achieve? I expect to see formal theorems and rigorous proofs.
>
> **A**
> In the previous manuscript, we only show the analysis that the optimal policy trained with the expected quadratic utility function is mean-variance efficient. This is because while EQUMRL has obviously such property in asymptotic behavior, it is not tractable to derive non-asymptotic results without imposing some restrictions to algorithms.
>
> **In Theorem 1 of the updated manuscript, we show the local convergence of our proposed REINFORCE-based algorithm based on [1]**.
>
> Regarding the convergence rate and regret, if we restrict the class of algorithms, we could analyze EQUM by using the results in [1, 2]. However, to the best of our knowledge, the non-asymptotic analysis still has many open problems, and discussing this in our work may not be appropriate because we aim to establish a general framework. Thus, although non-asymptotic analysis is important for future work, it is not the scope of this manuscript.
>
> [1] Dimitri P. Bertsekas, John N. Tsitsiklis, "Neuro-dynamic programming", 1996
> [2] Alekh Agarwal, Sham M. Kakade, Jason D. Lee, Gaurav Mahajan, "On the Theory of Policy Gradient Methods: Optimality, Approximation, and Distribution Shift", arXiv: 1908.00261.
> [3] Junzi Zhang, Jongho Kim, Brendan O'Donoghue, Stephen Boyd, "Sample Efficient Reinforcement Learning with REINFORCE", AAAI 2021.

---

> > ### Comment · Reviewer_wFpo · 2021-11-21
> > **Comments after the Author's Response**
> >
> > Thank you for your response and the updated theorem.
> >
> > As you mentioned, could you show the regret result under reasonable restrictions of the class of your algorithms? (Asymptotic result is OK).
> >
> > Under the mean-variance objective, the optimal policy $\pi^*(s)$ will depend on the historical trajectory instead of only the current state $s$. Could you explain the intuition on why/how your algorithm can converge to the non-stationary optimal policy? Is it simply because you consider parameterized policy and use policy gradient descent?

---

> > > ### Author Response · Authors · 2021-11-22
> > > **Re: Comments after the Author's Response**
> > >
> > > We thank the reviewer again for the reply. Our replies to your comments are listed below.
> > >
> > > **Q1**
> > > As you mentioned, could you show the regret result under reasonable restrictions of the class of your algorithms? (Asymptotic result is OK).
> > >
> > > **A1**
> > > For instance, as we explain in the updated manuscript, Zhang et al. (2021) considers "classical policy gradient methods that compute an approximate gradient with a single trajectory or a fixed size mini-batch of trajectories under soft-max parametrization and log-barrier regularization,
> > > along with the widely-used REINFORCE gradient estimation procedure" and provides "global convergence and sample efficiency results for the well-known REINFORCE algorithm." In addition, the paper assumes that a model is a fully tabular setting (finite state and action spaces). Thus, if we restrict the class of the policy and algorithm, we can derive global convergence and regret. However, the empirical performance is considered to be better with more sophisticated parametric models like neural networks, and we do not use such an algorithm used in Zhang et al. (2021) in our experiments. Therefore, it is difficult to present the regret result in our manuscript.
> > >
> > > On the other hand, as Zhang et al. (2021) has just been accepted for AAAI 2021, a more general result that does not restrict the policy and algorithm is an open problem for the RL community. Therefore, its generalization is outside the scope of our work on mean-variance RL.
> > >
> > > Zhang, J., Kim, J., O'Donoghue, B., and Boyd, S. Sample efficient reinforcement learning with rein-force. AAAI 2021
> > >
> > > **Q2**
> > > Under the mean-variance objective, the optimal policy $\pi*(s)$ will depend on the historical trajectory instead of only the current state $s$. Could you explain the intuition on why/how your algorithm can converge to the non-stationary optimal policy? Is it simply because you consider parameterized policy and use policy gradient descent?
> > >
> > > **A2**
> > > Yes, the optimal policy of EQUM will depend on the past reward sequence (more specifically, the past cumulative reward) if the feature vector of state observation at each time step includes the past cumulative reward. We call this past cumulative reward the reward feature. If you regard the state as the feature vector without the reward feature, the optimal policy in the EQUM may look like a non-Markov policy. However, if we regard the state as the feature vector with the reward feature, then the optimal policy in the EQUM is a Markov policy. It is common in RL that each element of the state feature vector depends on past trajectories, not only the reward feature.

---

> > > > ### Comment · Reviewer_wFpo · 2021-11-26
> > > > **Additional Comments**
> > > >
> > > > Thank you for your reply.
> > > >
> > > > To my best knowledge, in recent years, there are a number of papers that study regret bounds for (online) RL under tabular or linear MDP settings.
> > > >
> > > > For example,
> > > >
> > > > Azar, M. G., Osband, I. and Munos, R. (2017). Minimax regret bounds for reinforcement
> > > > learning. In Proceedings of the 34th International Conference on Machine Learning-Volume 70.
> > > > JMLR. org.
> > > >
> > > > Jin, C., Allen-Zhu, Z., Bubeck, S. and Jordan, M. I. (2018). Is Q-learning provably efficient?
> > > > In Advances in Neural Information Processing Systems.
> > > >
> > > > Jin, C., Yang, Z., Wang, Z. and Jordan, M. I. (2020). Provably efficient reinforcement learning with linear function approximation. In Conference on Learning Theory.
> > > >
> > > > I more or less understand that it is hard to analyze a policy gradient based RL algorithm with sophisticated parametric models like neural networks. But your reply did not show me sufficient contributions of this paper.
> > > >
> > > > In my opinion, this paper lacks solid theoretical guarantees, and its experiment part looks standard (does not provide empirical evaluations under sufficiently comprehensive and challenging experimental benchmarks/setups). I did not find enough reasons to recommend accept.
> > > > So I plan to stick to my score 5.

---

> > > > > ### Author Response · Authors · 2021-11-26
> > > > > **Re: Additional Comments**
> > > > >
> > > > > Thank you for sharing some interesting work!
> > > > >
> > > > > Unfortunately, even if there is a theoretical analysis **of the value iteration** in liner MDP, this does not mean that it is possible to analyze the gradient method with linear policy. In the state-of-the-art results, like Zhang et al. AAAI 2021. that we mentioned, the regret analysis is only shown for a more simplified case (e.g., a tabular policy in Zhang et al. 2021 and LQR in Jin et al. 2020), and is an open question for the RL community.
> > > > >
> > > > > Thus, it is **beyond the scope of our paper** to show regret analysis that the reviewer suggests as long as we use the policy gradient method (**with function approximation without the linear MDP assumption**). **However, we will add this clarification to our paper**.
> > > > >
> > > > > (For example, see Appendix E in Zhang et al. AAAI 2021 for recent studies on regret bounds)
> > > > > (We also note that the use of the policy gradient and experimental setting follow the existing literature in mean-variance RL)

---

### Official Review · Reviewer_FXvX · 2021-11-08

**Correctness:** 4
**Technical Novelty And Significance:** 3
**Empirical Novelty And Significance:** 2
**Recommendation:** 5
**Confidence:** 3

**Main Review:**

Strength:
The method in this paper is intuitive, and it builds a connection between utility optimization, a popular topic in finance, and reinforcement learning. The proposed method can mitigate the double sampling issue that appeared in prior work, and can be combined with any existing policy-gradient-based algorithms. The experimental results show that the algorithm has better performance compared to baselines.

Weakness:
I think there are some weakness in the experimental section. In Section 6.2, the portfolio management experiment using real-world data does not seem to be an RL problem. The dataset is static, which contains the returns of each asset at each time. This return does not change when the agent's action (weight w) changes. The policy network is essentially a "classifier" that maps the input feature vector (returns of assets) to a probability vector. Thus, the problem is more like an online learning (or supervised laerning) problem rather than an RL problem.

I agree that for this problem, it is still useful to better optimize the mean-variance trade-off, but I think the authors should clarify this better to avoid confusion. Moreover, since the authors propose this algorithm as an RL algorithm, it would be interesting to test the performance of the algorithm on a more realistic RL benchmark.

Minor:
Section 2.1: “A typical method for consider the trade-off is train a policy under some constrains”: this sentence has grammar mistake and typo.
Section 6.2: “The policy determines e the weight.”: typo.


**Summary Of The Paper:**

This paper proposes a method for mean-variance trade-off optimization in RL. The method, named EQUMRL, tries to optimize the Pareto efficiency by maximizing the quadratic utility function. The proposed method mitigates the double sampling issue appeared in prior work thus simplifies the optimization procedure. The authors also provided experiments to demonstrate the benefits of the algorithm using both synthetic datasets and real world data.

**Summary Of The Review:**

The proposed algorithm is interesting and has solid contribution. The experiment section is not satisfactory enough.

---

> ### Author Response · Authors · 2021-11-19
> **Response to Reviewer FXvX**
>
> Thank you for your insightful comments.
> We have updated the manuscript by fixing the typos and adding more explanation on the experiments of the portfolio management problem.
>
> We answer the following question.
>
> **Q**
> I think there are some weakness in the experimental section. In Section 6.2, the portfolio management experiment using real-world data does not seem to be an RL problem. The dataset is static, which contains the returns of each asset at each time. This return does not change when the agent's action (weight w) changes. The policy network is essentially a "classifier" that maps the input feature vector (returns of assets) to a probability vector. Thus, the problem is more like an online learning (or supervised laerning) problem rather than an RL problem.
>
> I agree that for this problem, it is still useful to better optimize the mean-variance trade-off, but I think the authors should clarify this better to avoid confusion. Moreover, since the authors propose this algorithm as an RL algorithm, it would be interesting to test the performance of the algorithm on a more realistic RL benchmark.
>
> **A**
> In our portfolio management problem, it makes sense to consider a dynamic decision-making problem with RL for the following reasons:
> - there is a trading cost proportional to the change of the portfolio weight (change of the action) between two rounds as Tamar et al. (2012) and Xie et al. (2018); that is, a policy with frequent change of action will suffer from the trading cost;
> - **the per-step optimization does not imply the trajectory optimization**; that is, even if the expected utility function is optimized at each time, the trajectory expected utility function is not optimized (for example, when we incur the low variance at around, we can take high-risk high-return choice in the other periods).
> - To control the variance well, we include the empirical mean and variance until the period $t$ into the state $S_t$.
> Thus, our portfolio management experiment works as an RL problem.
>
> To clarify this, in the update manuscript, we add the following sentence in Section 4.1:
> > Unlike the expected cumulative reward maximization in the standard RL setting, at time $t$, it is desirable to include the past cumulative reward to the state $S_t$ because our objective function depends on it even given $S_t$. Let us consider the objective at time $t$ with the infinite horizon setting:
> $$\alpha E_{\pi_\theta}\left[G_{0:\infty} - \beta G^2_{0:\infty}\big| S_0, A_0, r(S_0, A_0), \dots, S_{t-1}, A_{t-1}, r(S_{t-1}, A_{t-1}), S_t\right]$$
> $$= C + \alpha\gamma^t((1-2\beta\gamma^tG_{0:t-1})E_{\pi_\theta}[G_{t:\infty}| S_t] - \beta \gamma^t E_{\pi_\theta}[G^2_{t:\infty}| S_t])$$
> where $C$ is a constant and recall that $G_{0:t-1} = \sum^{t-1}_{i=0}\gamma^i r(S_i, A_0)$.Thus, for a better decision-making, we include the past cumulative reward into the state space.
>
> Besides, in Section 6.2, we also add the following description on the MDP in the experiments using the real-world financial dataset:
> > Consider an episodic MDP. Let the action space be the set of $m$ assets, and $y_{a, t}$ and $w_{a,t}$ be the return and the portfolio weight of an asset $a$ at time $t$. The reward (portfolio return) at time $1 \leq t \leq T$ is defined as $y_t = \sum_{a=1}^m y_{a, t} w_{a, t} - \Lambda \sum_{a=1}^m|w_{a, t} - w_{a, t-1}|$, where $\Lambda = 0.001$ is the penalty of the portfolio weight turnover(change of the action). On the $t$-th period, the agent observes a state $S_t=((y_{a,t-1},...,y_{a,t-12}, w_{a,t-1}), \sum^{t-1}y_s)$, and decides a portfolio weight $(w_{a,t})$ as $w_{a,t}=\pi(a, s_t)$. Between periods $t$ and $t+1$, the agent has an asset $a$ with the ratio $w_{a,t}$.

---

> > ### Comment · Reviewer_FXvX · 2021-11-30
> > **After response**
> >
> > Thanks for your response. I agree with you that in this problem, it is the cumulative reward (trajectory) that matters, rather than a single-step reward, because of the switching cost. In this case, I agree that we can apply an RL algorithm in this setting.
> >
> > However, since for each step, the returns of all the assets are revealed to the agent, so the problem seems to be more similar to an adaptive online learning problem or an online learning problem with switching cost. Some examples are:
> >
> > Cesa-Bianchi et al., Online Learning with Switching Costs and Other Adaptive Adversaries.
> >
> > Rangi and Franceschetti, Online learning with feedback graphs and switching costs.

---

> > > ### Author Response · Authors · 2021-12-06
> > > **Re: After response**
> > >
> > > Thank you for sharing the interesting papers. They are online learning, not RL; that is, their setting is different from ours. e.g., they consider an adversary with switching costs or problem setting different from MDPs. On the other hand, our problem formulation is RL on MDP by definition.
> > >
> > > **Note that online learning usually assumes a non-stationary environment under interaction with an adversarial agent while the MDP considers a stationary environment.**
> > >
> > > Besides, as we explained, to consider the mean-variance trade-off, we need to incorporate the past cumulative reward in the state; that is, the methods of the shared papers cannot solve the problem. This problem setting is accepted in existing mean-variance RL (Bisi et al. 2020) and RL applications in finance (Deng et al. 2016).
> > >
> > > Yue Deng, Feng Bao, Youyong Kong, Zhiquan Ren, and Qionghai Dai. Deep Direct Reinforcement Learning for Financial Signal Representation and Trading. IEEE2016.

---

### Author Response · Authors · 2021-11-26
**To All Reviewers**

Thank you to all reviewers for your insightful comments.

The followings are summaries of our clarifications and major revisions in the updated manuscript:
1. We added the pseudo-code of the REINFORCE-based algorithm.
2. We added a new experimental result of the per-step setting.
3. A reviewer asks us whether the experimental is not well-defined as a task of RL. To clarify this, we explicitly described the MDP of the problem. In our problem, there are trading costs and variance in the objective. For these two terms, considering dynamic-decision making is meaningful; that is, it provides a better answer than considering only instantaneous decision-making.
4. A reviewer suggests adding theoretical analysis. In the updated manuscript, we added the local convergence of the REINFORCE-based algorithm (Theorem 1). On the other hand, obtaining regret analysis in an online setting is still an open problem in the RL community. For instance, even a recent theoretical paper reports the regret analysis for a tabular policy trained with the log-barrier regularization term (Zhang et al. 2021). Such a setting is different from ours, and showing theoretical results under the setting is not the scope of our work. This problem should be discussed in another future paper in general RL, not our manuscript in mean-variance RL.
5. A reviewer comments that we need more clarification on why our proposed method outperforms the existing methods. In our first draft, e explain them in several parts, such as Remark 1. One of the algorithms seems to have a technical issue, and emphasizing the issue would be a discussion of the soundness of that paper. Although we briefly mention the issue, we did not pursue it because such a debate would not be appropriate to be included.

Zhang, J., Kim, J., O’Donoghue, B., and Boyd, S. Sample efficient reinforcement learning with rein-force. AAAI 2021

---

### Decision · Program_Chairs · 2022-01-20

**Decision:**

Reject

**Comment:**

This is a borderline paper with some reviewers voted for acceptance and some think it is not still ready. What is clear is more efforts by the authors is needed to make the paper appealing to reviewers with different interests. Changes such as better writing, more in depth literature review, more convincing experiments can definitely improve the quality of the paper. I personally do not think regret analysis is needed for this work, but it was mentioned by a reviewer. I would suggest the authors to use the reviewers' comments, revise their work, and prepare it for future conferences.